# Transcriptional cartography integrates multiscale biology of the human cortex

Konrad Wagstyl[1]*, Sophie Adler[2], Jakob Seidlitz[3,4], Simon Vandekar[5], Travis T Mallard[6,7], Richard Dear[8], Alex R DeCasien[9], Theodore D Satterthwaite[3,10], Siyuan Liu[9], Petra E Vértes[8], Russell T Shinohara[11], Aaron Alexander-Bloch[3,4], Daniel H Geschwind[12], Armin Raznahan[9]

[1]Wellcome Centre for Human Neuroimaging, University College London, London, United Kingdom; [2]UCL Great Ormond Street Institute for Child Health, Holborn, United Kingdom; [3]Department of Psychiatry, University of Pennsylvania, Philadelphia, United States; [4]Department of Child and Adolescent Psychiatry and Behavioral Science, The Children's Hospital of Philadelphia, Philadelphia, United States; [5]Department of Biostatistics, Vanderbilt University, Nashville, United States; [6]Psychiatric and Neurodevelopmental Genetics Unit, Center for Genomic Medicine, Massachusetts General Hospital, Boston, United States; [7]Department of Psychiatry, Harvard Medical School, Boston, United States; [8]Department of Psychiatry, University of Cambridge, Cambridge, United Kingdom; [9]Section on Developmental Neurogenomics, Human Genetics Branch, National Institute of Mental Health, Bethesda, United States; [10]Lifespan Informatics and Neuroimaging Center, University of Pennsylvania School of Medicine, Philadelphia, United States; [11]Penn Statistics in Imaging and Visualization Center, Department of Biostatistics, Epidemiology and Informatics, Perelman School of Medicine, University of Pennsylvania, Philadelphia, United States; [12]Center for Autism Research and Treatment, Semel Institute, Program in Neurogenetics, Department of Neurology and Department of Human Genetics, David Geffen School of Medicine, University of California, Los Angeles, Los Angeles, United States

*For correspondence:
k.wagstyl@ucl.ac.uk

**Abstract** The cerebral cortex underlies many of our unique strengths and vulnerabilities, but efforts to understand human cortical organization are challenged by reliance on incompatible measurement methods at different spatial scales. Macroscale features such as cortical folding and functional activation are accessed through spatially dense neuroimaging maps, whereas microscale cellular and molecular features are typically measured with sparse postmortem sampling. Here, we integrate these distinct windows on brain organization by building upon existing postmortem data to impute, validate, and analyze a library of spatially dense neuroimaging-like maps of human cortical gene expression. These maps allow spatially unbiased discovery of cortical zones with extreme transcriptional profiles or unusually rapid transcriptional change which index distinct micro-structure and predict neuroimaging measures of cortical folding and functional activation. Modules of spatially coexpressed genes define a family of canonical expression maps that integrate diverse spatial scales and temporal epochs of human brain organization – ranging from protein–protein interactions to large-scale systems for cognitive processing. These module maps also parse neuro-psychiatric risk genes into subsets which tag distinct cyto-laminar features and differentially predict the location of altered cortical anatomy and gene expression in patients. Taken together, the methods, resources, and findings described here advance our understanding of human cortical organization and offer flexible bridges to connect scientific fields operating at different spatial scales of human brain research.

## eLife assessment

This study provides continuous maps of human brain gene expression and explores their relationship with a large variety of microscopic and macroscopic aspects of brain organisation. The authors provide **convincing** evidence for a relationship between gene expression maps with various aspects of the anatomy of adult brains, during development, and in the case of mental disorders. The data and methods introduced can be an **important** tool for neuroimaging research.

## Introduction

The human cerebral cortex is an astoundingly complex structure that underpins many of our distinctive facilities and vulnerabilities (*Geschwind and Rakic, 2013*). Achieving a mechanistic understanding of cortical organization in health and disease requires integrating information across its many spatial scales: from macroscale cortical folds and functional networks (*Glasser et al., 2016*) to the gene expression programs that reflect microscale cellular and laminar features (*Hawrylycz et al., 2012*; *Kelley et al., 2018*). However, a hard obstacle to this goal is that our measures of the human cortex at macro- and microscales are fundamentally mismatched in their spatial sampling. Macroscale measures from in vivo neuroimaging provide spatially dense estimates of structure and function, but microscale measures of gene expression are gathered from spatial discontinuous postmortem samples that have so far only been linked to macroscale features using methodologically imposed cortical parcellations (*Hansen et al., 2021*; *Larivière et al., 2021*; *Seidlitz et al., 2020*). Consequently, local transitions in human cortical gene expression remain uncharacterized and unintegrated with the spatially fine-grained topographies of human cortical structure and function that are revealed by in vivo neuroimaging (*Gryglewski et al., 2018*; *Markello et al., 2021*). Finding a way to bridge this gap would not only enrich both our micro- and macroscale models of human cortical organization, but also provide an essential framework for translation across traditionally siloed scales of neuroscientific research.

Here, we use spatially sparse postmortem data from the Allen Human Brain Atlas (AHBA; *Hawrylycz et al., 2012*) to generate spatially dense cortical expression maps (DEMs) for 20,781 genes in the adult brain, with accompanying DEM reproducibility scores to facilitate wider usage. These maps allow a fine-grained transcriptional cartography of the human cortex, which we integrate with diverse genomic, histological, and neuroimaging resources to shed new light on several fundamental aspects of human cortical organization in health and disease. First, we show that DEMs can recover canonical gene expression boundaries from in situ hybridization (ISH) data, predict previously unknown expression boundaries, and align with regional differences in cortical organization from several independent data modalities. Second, by focusing on the local transitions in gene expression which are captured by DEMs, we reveal a close spatial coordination between molecular and functional specializations of the cortex and establish that the spatial orientation of cortical folding and function at macroscale is aligned with local tangential transitions in cortical gene expression. Third, by defining and annotating gene co-expression modules across the cortex at multiple scales we systematically link macroscale measures of cortical structure and function in vivo to postmortem markers of cortical lamination, cellular composition, and development from early fetal to late adult life. Finally, as a proof of principle, we use this novel framework to secure a newly integrated multiscale understanding of atypical brain development in autism spectrum disorder (ASD).

The tools and results from this analysis of the human cortex, which we collectively call Multiscale Atlas of Gene expression for Integrative Cortical Cartography (MAGICC), open up an empirical bridge that can now be used to connect cortical models (and scientists) that have so far operated at segregated spatial scales. To this end, we share (i) all gene-level DEMs and derived transcriptional landscapes in neuroimaging-compatible files for easy integration with in vivo macroscale measures of human cortical structure and function; and (ii) all gene sets defining spatial subcomponents of cortical transcription for easy integration with any desired genomic annotation (https://github.com/kwagstyl/magicc).

## Results

### Creating and benchmarking spatially dense maps of human cortical gene expression

To create a dense transcriptomic atlas of the cortex, we used AHBA microarray measures of gene expression for 20,781 genes in each of 1304 cortical samples from six donor left cortical hemispheres ('Materials and methods,'Table S1 *Supplementary file 1*). We extracted a model of each donor's cortical sheet by processing their brain MRI scan and identified the surface location (henceforth 'vertex') of each postmortem cortical sample in this sheet ('Materials and methods,' *Figure 1a*). For each gene, we then propagated measured expression values into neighboring vertices using nearest-neighbor interpolation followed by smoothing ('Materials and methods,' *Figure 1b and c*). Expression values were scaled across vertices and these vertex-level expression maps were averaged across donors to yield a single DEM for each gene, which provided estimates of expression at ~30,000 vertices across the cortical sheet (e.g., DEM for PVALB, upper panel of *Figure 1d*). These fine-grained vertex-level expression measures also enabled us to estimate the orientation and magnitude of expression change for each gene at every vertex (e.g., dense expression change map for PVALB, lower panel of *Figure 1d*).

We assessed the reproducibility of DEMs by repeating the above process (*Figure 1*) after repeatedly splitting the donors into non-overlapping groups of varying size and using learning curve analyses to estimate the DEM reproducibility achieved by our full set of six donors. For cortically expressed genes ('Materials and methods,' *Supplementary file 2*), the average reproducibility of gene expression maps was $r_{gene}$ = 0.58 (correlation of expression values for a gene across vertices), and the average reproducibility of ranked gene expression at each vertex was $r_{vertex}$ = 0.63 (correlation of expression values at a vertex across genes) (*Figure 1—figure supplement 1c-d*). These estimates were both substantially lower for genes not reported to be cortically expressed in the independent Human Protein Atlas ($r_{gene}$ = 0.34, $t$ = 37.6, p<0.001 and $r_{vertex}$ = 0.39, $t$ = 273.6, p<0.001, respectively, 'Materials and methods,' *Supplementary file 2*). Genes without recorded cortical expression were threefold enriched (p=0) among the 9647 genes with estimated DEM reproducibility values of $r$ < 0.5. Regional differences in the density of postmortem sampling in the AHBA did not influence DEM reproducibility or the magnitude of local expression change captured by DEMs ('Materials and methods,' *Figure 1—figure supplement 1h*). Thus, remedying the current lack of any spatially dense gene expression maps in the human cortex, we provide DEMs (and accompanying dense expression change maps) for 20,781 genes and establish that >11k of these DEMs show a spatial reproducibility score of $r_{gene}$ > 0.5 between sets of unrelated individuals. Gene-level DEM reproducibility scores allow future users to filter on this feature as desired, and we establish that key analytic outputs from DEMs (see below) show good reproducibility between unrelated individuals and can be recovered at different DEM reproducibility filters.

Given that DEMs were generated by interpolating expression values between sampled regions, we assessed whether DEMs could recover sharp local microscale transitions in gene expression that could theoretically be obscured by interpolation. Of the very few such transitions that have been verified by ISH in humans, the best established occurs between occipital areas V1 and V2 (*Zeng et al., 2012*). All four genes known to show a sharp V1/V2 expression boundary across layers by ISH – SYT6, TLE4, PCP4, PENK – exhibited qualitatively and quantitatively sharp expression transitions at the V1/V2 boundary in their DEMs (*Figure 1e*, *Figure 1—figure supplement 2a-d*). Motivated by this validation, we next asked whether DEMs could identify previously unknown expression boundary markers in the human cortex. To achieve this, we took advantage of extensive existing ISH data between parahippocampal (area PeEc) and fusiform gyri (area TF). We ranked genes by the magnitude of their expression gradient between these cortical regions in DEMs ('Materials and methods') and identified four genes with sharp expression transitions predicted by DEMs – NGB,HTR2A (TF > PeEc) and NTS, CHRNA3 (PeEc > TF) – for which independent ISH data were available. Expression profiling in ISH slabs verified the existence of sharp expression transition for all four genes (*Figure 1f*, *Figure 1—figure supplement 2e-g*). As the V1/V2 and the PeEc/TF boundaries both involve transitions between classical laminar types in cortical regions with highly conserved anatomical patterning (*von Economo and Koskinas, 1925*), we also tested whether DEMs could recover expression boundaries in more variable and uniformly laminated association cortex (*Ronan and Fletcher, 2015*). No such expression boundaries have been described in humans by ISH, but there are reports of sharp expression boundaries

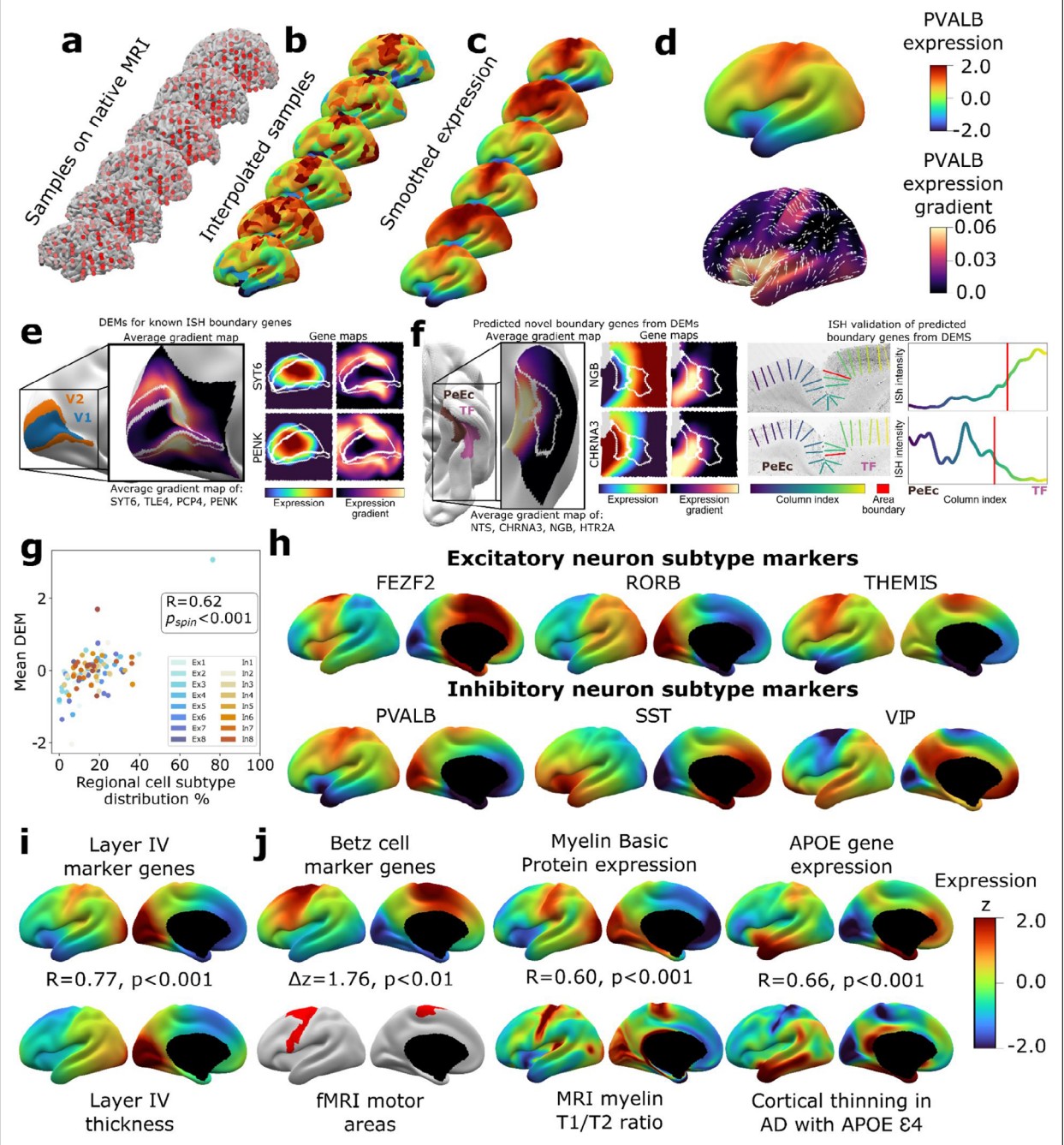

**Figure 1.** Creating and benchmarking spatial dense gene expression maps in the human cortex. (**a**) Spatially discontinuous Allen Human Brain Atlas (AHBA) microarray samples (red points) were aligned with MRI-derived cortical surface mesh reconstructions. (**b**) AHBA vertex expression values were propagated using nearest-neighbor interpolation and subsequently smoothed (**c**). (**d**) Subject-level maps were z-normalized and averaged to generate a single reference dense expression map (DEM) for each gene, as well as the associated expression gradient map (shown here for PVALB: top and bottom, respectively). (**e**) DEMs can recover known expression boundaries in in situ hybridization (ISH) data. Four canonical V1 area markers (***Zeng et al., 2012***) show a significantly sharp DEM expression gradient at the V1/V2 boundary (inset cortical map and ***Figure 1—figure supplement 2a, b***), which is also evident in all four individual gene DEMs and DEM gradients (SYT6, PENK, and ***Figure 1—figure supplement 2c***). (**f**) DEMs can discover previously unknown expression boundaries. Genes with high DEM gradients across the PeEc (parahippocampal) and TF (fusiform) gyri (inset cortical map) were validated in ISH data, showing sharp expression changes in both directions at this boundary (CHRNA3, NGB, and ***Figure 1—figure supplement 2d-f***). (**g**) Illustrative comparisons of selected DEMs against regional variation in microscale measures of cellular composition: scatterplot showing the global correlation of regional cellular proportions from single nucleus RNAseq (snRNAseq) across 16 cells and 6 regions (***Lake et al., 2016***) with DEM values for corresponding cell-type marker genes ($R = 0.48$, $p_{spin} < 0.001$, excluding Ex3-V1 and In8-BA10 outlier samples). (**h**) DEMs for markers of six neuronal subtypes (three excitatory: FEZF2, RORB, THEMIS; three inhibitory: PVALB, SST, VIP) based on recently validated subtype marker genes (***Bakken et al.,***

*Figure 1 continued on next page*

*Figure 1 continued*

*2021*; *Hodge et al., 2019*). (**i**) Illustrative comparison of layer IV marker DEMs with corresponding mesoscale cortical measure of layer IV thickness from a 20 μm 3D histological atlas of cortical layers. (**j**) Illustrative comparisons of selected DEMs with corresponding macroscale cortical measures from independent neuroimaging markers.

The online version of this article includes the following figure supplement(s) for figure 1:

**Figure supplement 1.** Reproducibility of Dense Expression Maps (DEMs) interpolated from spatially sparse postmortem measures of cortical gene expression.

**Figure supplement 2.** Validating and discovering area marker genes.

between frontal areas 44 and 45b for several genes in non-human primates: SCN1B, KCNS1, TRIM55 (*Chen et al., 2022*). These genes also exhibited high DEM gradients at the boundary between human frontal areas 44 and 45 (*Figure 1—figure supplement 2h-g*). Taken together, these observations demonstrate the capacity of DEMs to resolve sharp expression transitions and indicate that DEMs can be used to help target prospective postmortem validation of new expression boundaries in humans.

To benchmark and illustrate the use of DEMs to capture cortical features across contrasting spatial scales, we drew on selected micro- and macroscale cortical measures that DEMs should align with based on known biological processes (*Figure 1g–j*, 'Materials and methods'). To assess whether DEMs could recover microscale differences in cellular patterning across the cortical sheet, we considered the ground truth of neuronal cell-type proportions as measured by single-nucleus RNAseq (snRNAseq) across six different cortical regions (*Lake et al., 2016*). We observed a strong spatial correlation ($r = 0.6$, $p_{spin}<0.001$) between regional marker gene expression in DEMs and regional proportions of their corresponding neuronal subtypes from snRNAseq (*Figure 1g*, 'Materials and methods'). *Figure 1h* shows example marker gene DEMs for six canonical neuronal subtypes: three excitatory (FEZF2, RORB, THEMIS) and three inhibitory (PVAL, SST, VIP) (*Bakken et al., 2021*; *Hodge et al., 2019*). Next, to assess whether DEMs could recover regional variation in the mesoscale feature of cortical layering, we tested and verified that regional variation in the average DEM for layer IV marker genes (*He et al., 2017*; *Maynard et al., 2021*; *Zeng et al., 2012*) was highly correlated with regional variation in layer IV thickness as determined from a 3D histological atlas of cortical layers (*Wagstyl et al., 2020*; *Figure 1i*). Finally, we asked whether DEMs could recover spatially dense measures of regional variation across the cortical sheet as provided by neuroimaging data and found that maps from diverse measurement modalities showed strong and statistically significant spatial correlations with their corresponding DEM(s) relative to a null distribution based on random 'spinning' of maps (*Alexander-Bloch et al., 2018*; *Figure 1j*, 'Materials and methods,' all $p_{spin}<0.01$): (i) areas of cortex activated during motor fMRI tasks in humans (*Glasser et al., 2016*) vs. the average DEM for canonical cell markers of large pyramidal neurons (Betz cells) found in layer V of the motor cortex that are the outflow for motor movements (*Bakken et al., 2021*), (ii) an in vivo neuroimaging marker of cortical myelination (T1/T2 ratio [*Glasser and Van Essen, 2011*]) vs. the Myelin Basic Protein DEM, which marks myelin, and (iii) the degree of in vivo regional cortical thinning by MRI in Alzheimer's disease (AD) patients who have at least one APOE E4 variant (*Gutiérrez-Galve et al., 2009*; *LaMontagne et al., 2019*) vs. the APOE DEM (thinning map generated from 119 APOE E4 patients and 633 controls structural MRI [sMRI] scans as detailed in 'Materials and methods'), testing the hypothesis that higher regional APOE expression will result in greater cortical atrophy in individuals with the APOE E4 risk allele. Collectively, the above tests of reproducibility (*Figure 1—figure supplement 1*) and convergent validity (*Figure 1e–j*) supported the use of DEMs for downstream analyses.

## Defining and surveying the human cortex as a continuous transcriptional terrain

As an initial summary view of transcriptional patterning in the human cortex, we first averaged all 20,781 DEMs to represent the cortex as a single continuous transcriptional terrain, where altitude encodes the transcriptional distinctiveness (TD) of each cortical point (vertex) relative to all others (TD = mean(abs($z_{exp}$)), *Figure 2a*, *Video 1*). This terrain view revealed six statistically significant TD peaks ('Materials and methods,' *Figure 2a and b*) which recover all major archetypal classes of the mammalian cortex as defined by classical studies of laminar and myelo-architecture, connectivity, and functional specialization (*Mesulam, 1998*) encompassing primary visual (V1), somatosensory (Brodmann

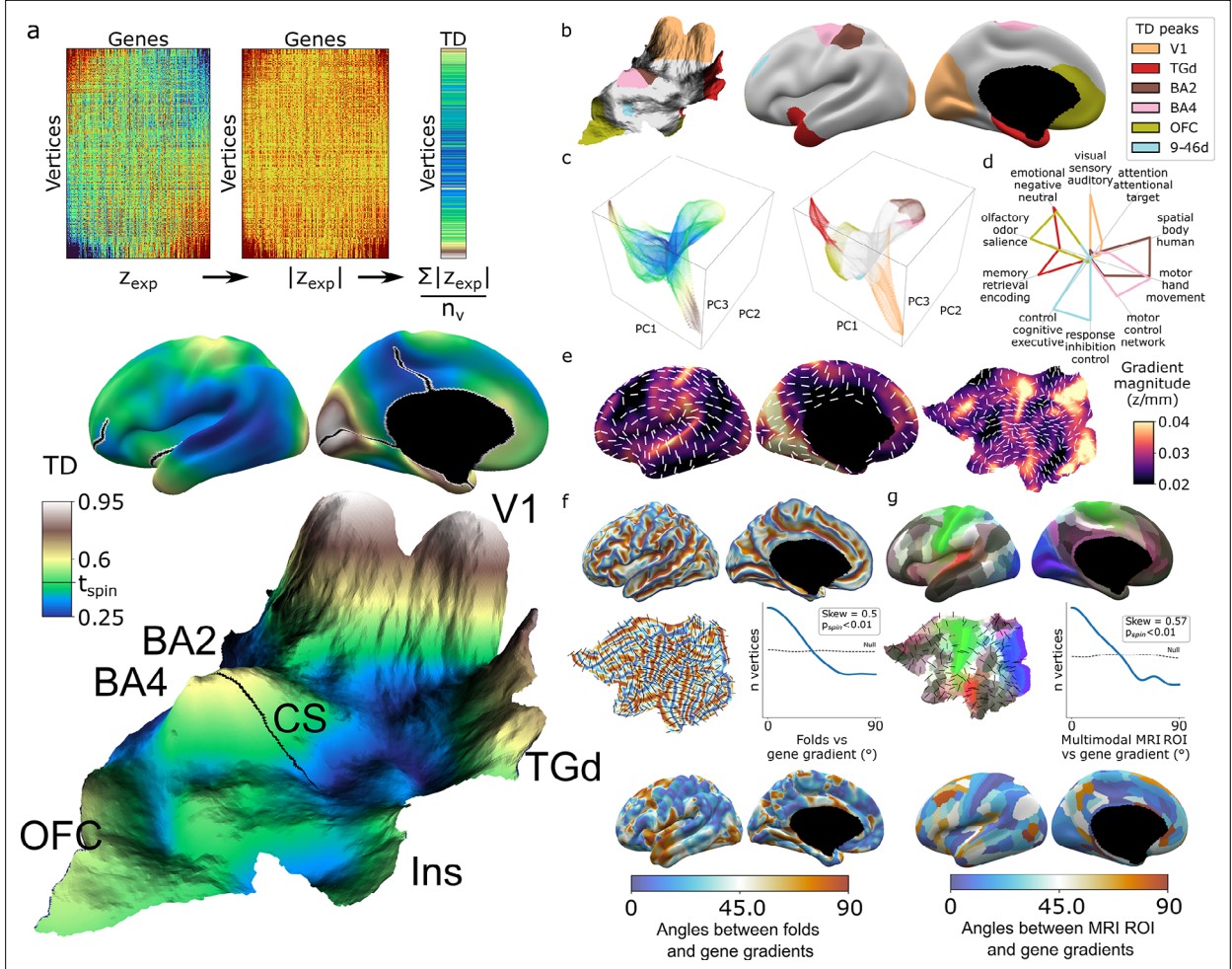

**Figure 2.** Mapping transcriptional distinctiveness (TD) in the human cortex and its alignment with macroscale structure and function. (**a**) Regional TD can be quantified as the mean absolute z-score of dense expression map (DEM) values at each vertex (top) and visualized as a continuous cortical map (middle, TD encoded by color) or in a relief map of the flattened cortical sheet (bottom, TD encoded by color and elevation, *Video 1*). Black lines on the inflated view identify cuts for the flattening procedure. The cortical relief map is annotated to show the central sulcus (CS), and peaks of TD overlying dorsal sensory and motor cortices (Brodmann areas, BA2, BA4), the primary visual cortex (V1), temporal pole (TGd), insula (Ins), and ventromedial prefrontal cortex (OFC). (**b**) Thresholding the TD map through spatial permutation of DEMs ($t_{spin}$ ; 'Materials and methods') and clustering significant vertices by their expression profile defined six TD peaks in the adult human cortex (depicted as colored regions on terrain and inflated cortical surfaces). (**c**) Cortical vertices projected into a 3D coordinate system defined by the first three principal components (PCs) of gene expression, colored by the continuous TD metric (left) and TD peaks (right). TD peaks are focal anchors of cortex-wide expression PCs. (**d**) TD peaks show statistically significant functional specializations in a meta-analysis of in vivo functional MRI data. (**e**) The average magnitude of local expression transitions across genes (color) and principal orientation of these transitions (white bars) varies across the cortex. (**f**) Cortical folds in Allen Human Brain Atlas (AHBA) donors (top surface maps and middle flat map) tend to be aligned with the principal orientation of TD change across cortical vertices (p<0.01, middle histogram, sulci running perpendicular to TD change), and the strength of this alignment varies between cortical regions. (**g**) Putative cortical areas defined by a multimodal in vivo MRI parcellation of the human cortex (*Glasser et al., 2016*) (top surface maps and middle flat map) also tend to be aligned with the principal direction of gene expression change across cortical vertices (p<0.01, middle histogram, sulci running perpendicular to long axis of area boundaries), and the strength of this alignment varies between cortical areas.

The online version of this article includes the following figure supplement(s) for figure 2:

**Figure supplement 1.** Characterizing bulk transcriptome.

area [BA] [*Brodmann, 1909*] 2), and motor cortex (BA 4), as well as limbic (temporal pole centered on dorsal temporal area G [TGd]; *von Economo and Koskinas, 1925*, ventral frontal centered in orbitofrontal cortex [OFC]) and heteromodal association cortex (BA 9-46d). Of note, our agnostic parcellation of all TD peak vertices by their ranked gene lists ('Materials and methods') perfectly cleaved BA2 and BA4 along the central sulcus, despite there being no representation of this macroanatomical

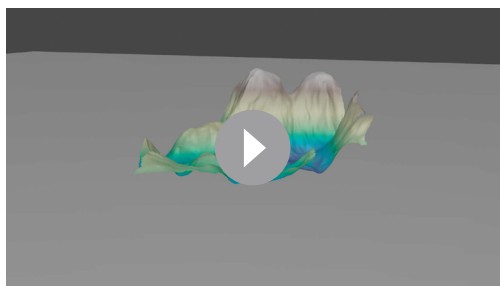

**Video 1.** Visualisation of Transcriptional Distinctiveness (TD) in the human cortex, encoded by both color and elevation.
https://elifesciences.org/articles/86933/figures#video1

landmark in DEMs. The TD map observed from the full DEMs library was highly stable between all disjoint triplets of donors ('Materials and methods,' *Figure 2—figure supplement 1a*, median cross-vertex correlation in TD scores between triplets $r = 0.77$) and across library subsets at all deciles of DEM reproducibility ('Materials and methods,' *Figure 2—figure supplement 1b*, cross-vertex correlation in TD scores $r > 0.8$ for the 3rd to 10th deciles), but was not recapitulated in spun null datasets (*Figure 2—figure supplement 1c*).

Integration with principal component analysis (PCA) of DEMs across vertices ('Materials and methods,' *Figure 2—figure supplement 1d and e*) showed that TD peaks constitute sharp poles of more recently recognized cortical expression gradients (*Burt et al., 2018*; *Figure 2c*). The 'area-like' nature of these TD peaks is reflected by the steep slopes of transcriptional change surrounding them (*Figure 2a and e*) and could be quantified as TD peaks being transcriptomically more distinctive than their physical distance from other cortical regions would predict (*Figure 2—figure supplement 1f and g*). In contrast, transitions in gene expression are more gradual and lack such sharp transitions in the cortical regions between TD peaks (*Figure 2a, c and e*, *Figure 2—figure supplement 1j*). Thus, because DEMs provide spatially fine-grained estimates of cortical expression and expression change, they offer an objective framework for arbitrating between area-based and gradient-based views of cortical organization in a regionally specific manner.

The TD peaks defined above exist as both discrete patches of cortex and the distinctive profile of gene expression which defines each peak, and this duality offers an initial bridge between macro- and microscale views of cortical organization. Specifically, we found that each TD peak overlapped with a functionally specialized cortical region based on meta-analysis of in vivo functional neuroimaging data (*Yarkoni et al., 2011*; 'Materials and methods,' *Figure 2d*, *Supplementary file 3*), and featured a gene expression signature that was preferentially enriched for a distinct set of biological processes, cell-type signatures, and cellular compartments ('Materials and methods,' *Supplementary file 2*). For example, the peaks overlapping area TGd and OFC were enriched for synapse-related terms, while BA2 and BA4 TD peaks were predominantly enriched for metabolic and mitochondrial terms. At a cellular level, V1 closely overlapped with DEMs for marker genes of the Ex3 neuronal subtype known to be localized to V1 (*Lake et al., 2016*), while BA4 closely overlapped Betz cell markers (*Bakken et al., 2021*; *Figure 2—figure supplement 1h*).

The expression profile of each TD peak was achieved through surrounding zones of rapid transcriptional change (*Figure 2a and e*, *Figure 2—figure supplement 1i and j*). We noted that these transition zones tended to overlap with cortical folds, suggesting an alignment between spatial orientations of gene expression and folding. To formally test this idea, we defined the dominant orientation of gene expression change at each vertex ('Materials and methods,' *Figure 2e*) and computed the angle between this and the orientation of folding ('Materials and methods'). The observed distribution of these angles across vertices was significantly skewed relative to a null based on random alignment between angles ($p_{spin} < 0.01$, *Figure 2f*, 'Materials and methods'), indicating that there is indeed a tendency for cortical sulci and the direction of fastest transcriptional change to run perpendicular to each other ($p_{spin} < 0.01$, *Figure 2f*). A similar alignment was seen when comparing gradients of transcriptional change with the spatial orientation of putative cortical areas defined by multimodal functional and structural in vivo neuroimaging (*Glasser et al., 2016*) (expression change running perpendicular to area long axis, $p_{spin} < 0.01$, *Figure 2g*, 'Materials and methods'). Visualizing these expression-folding and expression-areal alignments revealed greatest concordance over sensorimotor, medial occipital, cingulate, and posterior perisylvian cortices (with notable exceptions of transcription change running parallel to sulci and the long axis of putative cortical areas in lateral temporoparietal and temporopolar regions). As a preliminary probe for causality, we examined the developmental ordering of regional folding and regional transcriptional identity. Mapping the expression of high-ranking TD

genes in fetal cortical laser dissection microarray data (*Miller et al., 2014*) from 21 PCW (post conception weeks) ('Materials and methods') showed that the localized transcriptional identity of V1 and TGd regions in adulthood is apparent during the fetal periods that folding topology begins to emerge (*Chi et al., 1977*; *Xu et al., 2022*; *Figure 2—figure supplement 1k*). Thus, the unique capacity of DEMs to resolve local orientations of expression change reveals a close spatial alignment between regional transitions of cortical gene expression at microscale and regional transitions of cortical folding, structure, and function at macroscale.

## Cortical gene co-expression integrates diverse spatial scales of human brain organization

To complement the TD analyses above (*Figure 2*), we next used weighted gene co-expression network analysis (WGCNA; *Langfelder and Horvath, 2008*, 'Materials and methods', *Figure 3a*) to achieve a more systematic integration of macro- and macroscale cortical features. Briefly, WGCNA constructs a connectivity matrix by quantifying pairwise co-expression between genes, raising the correlations to a power (here 6) to emphasize strong correlations while penalizing weaker ones, and creating a topological overlap matrix (TOM) to capture both pairwise similarities expression and connectivity. Modules of highly interconnected genes are identified through hierarchical clustering. The resultant WGCNA modules enable topographic and genetic integration because they each exist as both (i) a single expression map (eigenmap) for spatial comparison with neuroimaging data (*Figure 3a and b*, 'Materials and methods') and (ii) a unique gene set for enrichment analysis against marker genes systematically capturing multiple scales of cortical organization, namely cortical layers, cell types, cell compartments, protein–protein interactions (PPI), and GO terms ('Materials and methods,' *Supplementary files 2 and 4*). Furthermore, whereas prior applications of WGCNA to AHBA data have revealed gene sets that covary in expression across many different compartments of the brain (*Hartl et al., 2021*; *Hawrylycz et al., 2015*; *Kelley et al., 2018*), using DEMs as input to WGCNA generates modules that are purely based on the fine-scale coordination of gene expression across the cortex. Using WGCNA, we identified 16 gene modules (M1–M16), which we then deeply annotated against independent measures of cortical organization at diverse spatial scales and developmental epochs (*Figure 3c*, 'Materials and methods'). Module eigenmaps were primarily driven by highly reproducible genes (*Figure 3—figure supplement 1a*) as were enrichments for annotational gene sets (median reproducibility of enriching genes = 0.59, p<0.001 elevated vs. background).

Several WGCNA modules showed statistically significant alignments with structural and functional features of the adult cerebral cortex from in vivo imaging ('Materials and methods,' *Figure 3c*; *Glasser and Van Essen, 2011*; *Yeo et al., 2011*). For example, (i) the M6 eigenmap was significantly positively correlated with in vivo measures of cortical thickness from sMRI and enriched within a limbic functional connectivity network defined by resting-state functional connectivity MRI, and (ii) the M8, M9, and M14 eigenmaps showed gradients of expression change that were significantly aligned with the orientation of cortical folding (especially around the central sulcus, medial prefrontal, and temporo-parietal cortices, *Figure 3—figure supplement 1b*). At microscale, several WGCNA module gene sets showed statistically significant enrichments for genes marking specific cortical layers (*He et al., 2017*; *Maynard et al., 2021*) and cell types (*Darmanis et al., 2015*; *Habib et al., 2017*; *Hodge et al., 2019*; *Lake et al., 2018*; *Lake et al., 2016*; *Li et al., 2018*; *Ruzicka et al., 2021*; *Velmeshev et al., 2019*; *Zhang et al., 2016*; 'Materials and methods,' *Figure 3c*, *Supplementary file 4*). These microscale enrichments were often congruent between cortical layers and cell classes annotations, and in keeping with the linked eigenmap (*Figure 3c*, *Supplementary file 4*). For example, M4, which was uniquely co-enriched for markers of endothelial cells and middle cortical layers, showed peak expression over dorsal motor cortices which are known to show expanded middle layers (*Bakken et al., 2021*; *Wagstyl et al., 2020*) with rich vascularization (*Pfeifer, 1940*) relative to other cortical regions. Similarly, M6, which was enriched for markers of astrocytes, microglia, and excitatory neurons, as well as layers 1/2, showed peak expression over rostral frontal and temporal cortices which are known to possess relatively expanded supragranular layers (*Wagstyl et al., 2020*) that predominantly contain the apical dendrites of excitatory neurons and supporting glial cells (*von Economo and Koskinas, 1925*). We also observed that modules with similar eigenmaps (*Figure 3—figure supplement 1c*), (including overlaps of multiple modules with the same TD peak) could show contrasting gene set enrichments. For example, M2 and M4 both showed peak expression of dorsal sensorimotor cortex

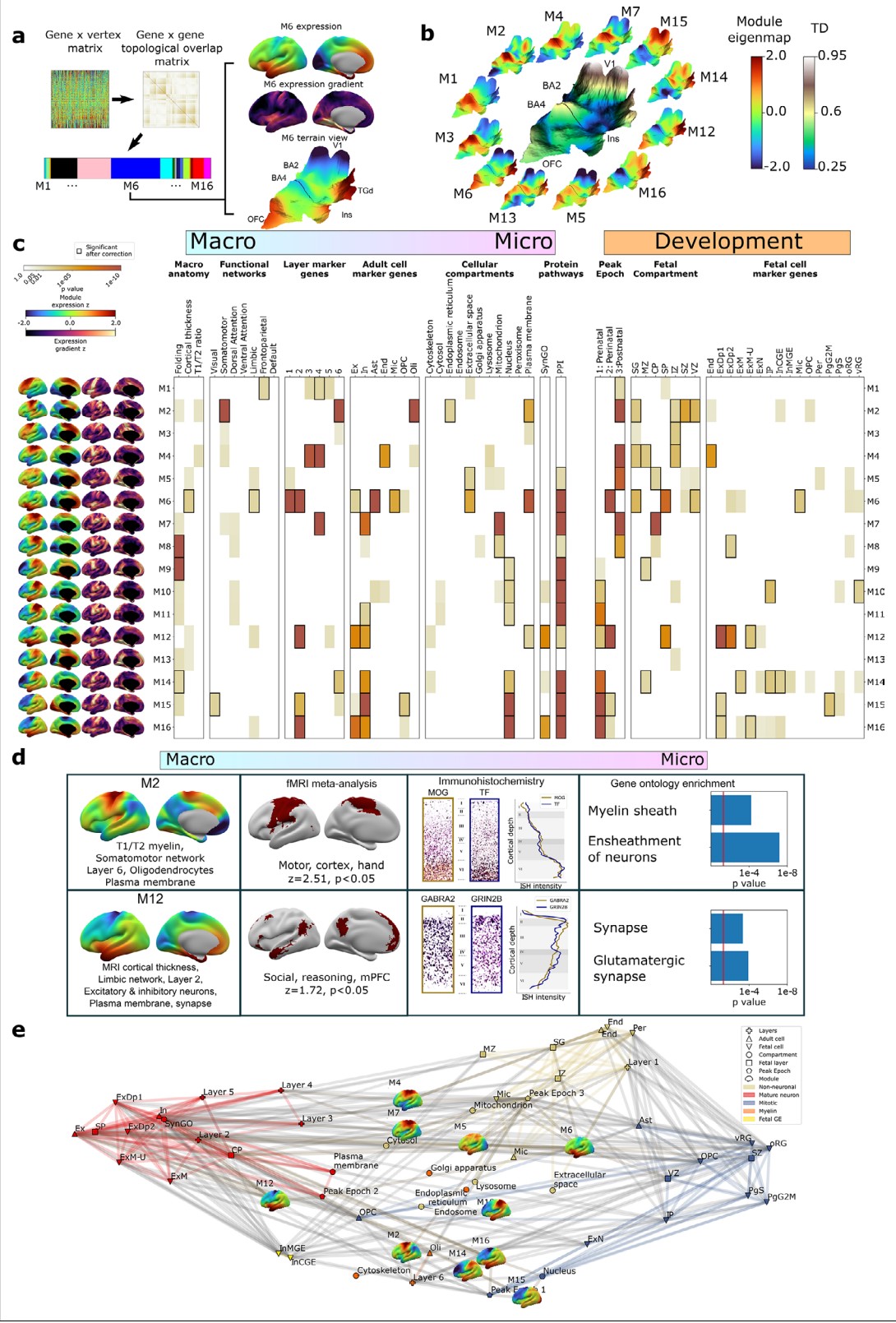

**Figure 3.** Cortex-wide gene co-expression patterns reflect multiple spatial scales and developmental epochs of brain organization. (**a**) Overview of weighted gene co-expression network analysis (WGCNA) pipeline applied to the full dense expression map (DEM) dataset. Starting top left: the pairwise DEM spatial correlation matrix is used to generate a topological overlap matrix between genes (middle top), which is then clustered. Of the 23 WGCNA-defined modules, 7 were significantly enriched for non-cortical genes and removed, leaving 16 modules. Each module is defined by a set

*Figure 3 continued on next page*

*Figure 3 continued*

of spatially co-expressed genes, for which the principal component of expression can be computed and mapped at each cortical point (eigenmap). M6 is shown as an example projected onto an inflated left hemisphere (M6 z-scored expression and M6 expression change), and the bulk transcriptional distinctiveness (TD) terrain view from *Figure 2* (M6 expression). (**b**) The extremes of WGCNA eigenmaps highlight different peaks in the cortical terrain: the main TD terrain colored by TD value (center, from *Figure 2*), surrounded by TD terrain projections of selected WGCNA eigenmaps. (**c**) WGCNA modules (eigenmaps and gradient maps, rows) are enriched for multiscale aspects of cortical organization (columns). Cell color intensity indicates pairwise statistical significance (p<0.05), while black outlines show significance after correction for multiple comparisons across modules. Columns capture key levels of cortical organization at different spatial scales (arranged from macro- to microscale) and developmental epochs: spatial alignment between module eigenmaps and in vivo MRI maps of cortical folding orientation, cortical thickness and T1/T2 ratio, fMRI resting-state functional networks; enrichment for module gene sets for independent annotations (*Supplementary file 2*) marking: cortical layers (*He et al., 2017*; *Maynard et al., 2021*); cell types (*Darmanis et al., 2015*; *Habib et al., 2017*; *Hodge et al., 2019*; *Lake et al., 2018*; *Lake et al., 2016*; *Li et al., 2018*; *Ruzicka et al., 2021*; *Velmeshev et al., 2019*; *Zhang et al., 2016*); subcellular compartments (*Binder et al., 2014*); synapse-related genes (*Koopmans et al., 2019*); protein–protein interactions between gene products (*Szklarczyk et al., 2019*); temporal epochs of peak expression (*Werling et al., 2020*) ('fetal': 8–24 21 post conception weeks [PCW]/''perinatal'' 24 PCW–6 mo/'postnatal' > 6 mo); transient layers of the mid-fetal human cortex at 21 PCW (*Miller et al., 2014*) (subpial granular zone [SG], marginal zone [MZ], cortical plate [CP], subplate [SP], intermediate zone [IZ], subventricular zone [SZ], and ventricular zone [VZ]); and fetal cell types at 17–18 PCW (*Polioudakis et al., 2019*). (**d**) Independent validation of multiscale enrichments for selected modules M2 and M12. M2 significantly overlaps the Neurosynth topic associated with the terms motor, cortex, and hand. Two high-ranking M2 genes, MOG and TF, exhibit clear layer VI peaks on in situ hybridization (ISH) and GO enrichment analysis myelin-related annotations. M12, overlapping the limbic network most closely overlapped the Neurosynth topic associated with social reasoning. Two high-ranking M22 genes GABRA2 and GRIN2B showed layer II ISH peaks and GO enrichment analysis revealed synaptic annotations. (**e**) Network visualization of pairwise overlaps between annotational gene sets used in (c), including WGCNA module gene sets (inset expression eigenmaps).

The online version of this article includes the following figure supplement(s) for figure 3:

**Figure supplement 1.** Characterization of WGCNA spatial, developmental and gene-set relationships.

(i.e., TD areas BA2 and BA4), but M2 captures a distinct architectonic signature of sensorimotor cortex from the mid-layer vascular signal of M4: expanded and heavily myelinated layer 6 (*Bakken et al., 2021*; *Palomero-Gallagher and Zilles, 2019*; *Wagstyl et al., 2020*; *Figure 3c*). The spatially co-expressed gene modules detected by WGCNA were not only congruently co-enriched for cortical layer and cell markers, but also for nanoscale features such as subcellular compartments (*Binder et al., 2014*; *Supplementary files 2 and 4*) (often aligning with the cellular enrichments) and PPIs (*Szklarczyk et al., 2019*; 'Materials and methods,' *Figure 3c*, *Supplementary file 4*). This demonstrates the capacity of our resource to tease apart subtle subcomponents of neurobiology based on cortex-wide expression patterns.

To further assess the robustness of these multiscale relationships, we focused on two modules with contrasting multiscale signatures – M2 and M12 – and tested for reproducibility of our primary findings (*Figure 3c*) using orthogonal methods. Our primary analyses indicated that M2 has an expression eigenmap which overlaps with the canonical somatomotor network from resting-state functional neuroimaging (*Yeo et al., 2011*) and contains genes that are preferentially expressed in cortical layer 6 from layer-resolved transcriptomics (*He et al., 2017*; *Maynard et al., 2021*), and in oligodendrocytes from snRNAseq (*Darmanis et al., 2015*; *Habib et al., 2017*; *Hodge et al., 2019*; *Lake et al., 2018*; *Lake et al., 2016*; *Li et al., 2018*; *Ruzicka et al., 2021*; *Velmeshev et al., 2019*; *Zhang et al., 2016*; *Figure 3c*). We were able to verify each of these observations through independent validations including spatial overlap of M2 expression with meta-analytic functional activations relating to motor tasks (*Yarkoni et al., 2011*); immunohistochemistry localization of high-ranking M2 genes to deep cortical layers (*Zeng et al., 2012*; 'Materials and methods'); and significant enrichment of M2 genes for myelin-related GO terms (*Figure 3d*, *Supplementary file 4*). By contrast, our primary analyses indicated that M12, which had peak expression over ventral frontal and temporal limbic cortices, was enriched for marker genes for layer 2, neurons and the synapse (*Figure 3c*). These multiscale enrichments were all supported by independent validation analyses, which showed that the M12 eigenmaps is enriched in a limbic network that is activated during social reasoning (*Yarkoni et al., 2011*) high-ranking M12 marker genes show elevated expression in upper cortical layers by immunohistochemistry (*Zeng et al., 2012*; 'Materials and methods'); and there is a statistically significant over-representation of synapse compartment GO terms in the M12 gene set (*Figure 3d*, *Supplementary file 4*).

## Linking spatial and developmental aspects of cortical organization

Given that adult cortical organization is a product of development, we next asked whether eigenmaps of adult cortical gene expression (*Figure 3a and b*) are related to the patterning of gene expression between fetal stages and adulthood. To achieve this, we tested WGCNA module gene sets for enrichment of developmental marker genes from three independent postmortem studies (rightmost columns, *Figure 3c*) capturing genes with differential expression between (i) three developmental epochs between 8 PCWs and adulthood (*BrainVar* dataset from prefrontal cortex [*Werling et al., 2020*]);(ii) seven histologically defined zones of mid-fetal (21 PCW) cortex (*Miller et al., 2014*; 'Materials and methods,' 2*Supplementary files 1 and 2*); and (iii) 16 mid-fetal (17–18 PCW) cell types (*Polioudakis et al., 2019*; 'Materials and methods,' *Supplementary file 2*).

Comparison with the *BrainVar* dataset revealed that most module eigenmaps (13 of all 16 cortical modules) were enriched for genes with dynamic, developmentally coordinated expression levels between early fetal and late adult stages (*Figure 3c*, *Supplementary file 4*). This finding was reinforced by supplementary analyses modeling developmental trajectories of eigenmap gene set expression between 12 PCW and 40 y in the *BrainSpan* dataset (*Li et al., 2018*; 'Materials and methods,' *Figure 3—figure supplement 1d*), and further qualified by the observation that several WGCNA modules were also differentially enriched for markers of mid-fetal cortical layers and cell types (*Miller et al., 2014*; *Polioudakis et al., 2019*; *Figure 3c*, *Supplementary file 4*). As observed for multiscale spatial enrichments (*Figure 3c and d*), the developmental enrichments of modules were often closely coordinated with one another, and eigenmaps with similar patterns of regional expression could possess different signatures of developmental enrichment. For example, the M6 and M12 eigenmaps shared a similar spatial expression pattern in the adult cortex (peak expression in medial prefrontal, anterior insula, and medioventral temporal pole), but captured different aspects of human brain development that aligned with the cyto-laminar enrichments of M6 and M12 in adulthood. The M6 gene set, which was enriched for predominantly glial elements of layers 1 and 2 in adult cortex, was also enriched for markers of mid-fetal microglia (*Polioudakis et al., 2019*), the transient fetal layers that are known to be particularly rich in mid-fetal microglia (subpial granular, subplate, and ventricular zone [*Monier et al., 2007*]), and the mid-late fetal epoch when most microglial colonization of the cortex is thought to be achieved (*Menassa and Gomez-Nicola, 2018*; *Figure 3c*). In contrast, the M12 gene set, which was enriched for predominantly neuronal elements of layer 2 in adult cortex, also showed enrichment for marker genes of developing fetal excitatory neurons, the fetal cortical subplate, and windows of mid-late fetal development when developing neurons are known to be migrating into a maximally expanded subplate (*Molnár et al., 2019*).

The striking co-enrichment of WGCNA modules for features of both the fetal and adult cortex (*Figure 3c*) implied a patterned sharing of marker genes between cyto-laminar features of the adult and fetal cortex. To more directly test this idea and characterize potential biological themes reflected by these shared marker genes, we carried out pairwise enrichment analyses between all annotational gene sets from *Figure 3c*. These gene sets collectively draw from a diverse array of study designs encompassing bulk, laminar, and single-cell transcriptomics of the human cortex between 10 PCW and 60 y of life ('Materials and methods'; *Darmanis et al., 2015*; *Habib et al., 2017*; *He et al., 2017*; *Li et al., 2018*; *Maynard et al., 2021*; *Miller et al., 2014*; *Polioudakis et al., 2019*; *Ruzicka et al., 2021*; *Velmeshev et al., 2019*; *Werling et al., 2020*; *Zhang et al., 2016*). Network visualization and clustering of the resulting adjacency matrix (*Figure 3—figure supplement 1e*) revealed an integrated annotational space defined by five coherent clusters (*Figure 3e*). A mature neuron cluster encompassed markers of postmitotic neurons and the compartments that house them in both fetal and adult cortex (red, *Figure 3e*, *Supplementary file 2*, example core genes: NRXN1, SYT1, CACNG8). This cluster also included genes with peak expression between late fetal and early postnatal life, and those localizing to the plasma membrane and synapse. A small neighboring fetal ganglionic eminence cluster (fetal GE, yellow, *Figure 3e*, *Supplementary file 2*, example core genes: NPAS3, DSX, DCLK2) contained marker sets for migrating inhibitory neurons from the medial and caudal ganglionic eminence in mid-fetal life. These two neuronal clusters – mature neuron and fetal GE – were most strongly connected to the M12 gene set ('Materials and methods'), which highlights medial prefrontal, and temporal cortices possessing a high ratio of neuropil:neuronal cell bodies (*Collins et al., 2010*; *Spocter et al., 2012*). A mitotic annotational cluster (blue, *Figure 3e*, *Supplementary file 2*, example core genes: CCND2, MEIS2, PHLDA1) was most distant from these two neuronal

clusters and included genes showing highest expression in early development as well as markers of cycling progenitor cells, radial glia, oligodendrocyte precursors, germinal zones of the fetal cortex, and the nucleus. This cluster was most strongly connected to the M15 gene set, which shows high expression over occipito-parietal cortices distinguished by a high cellular density and notably low expression in lateral prefrontal cortices, which possess low cellular density (*Collins et al., 2016*). The mature neuron and mitotic clusters were separated by two remaining annotational clusters for non-neuronal cell types and associated cortical layers. A myelin cluster (orange, *Figure 3e*, *Supplementary file 2*, example core genes: MOBP, CNP, ACER3) – which contained gene sets marking adult layer 6, oligodendrocytes, and organelles supporting the distinctive biochemistry and morphology of oligodendrocytes (the golgi apparatus, endoplasmic reticulum, and cytoskeleton) – was most connected to the M2 gene set highlighting heavily myelinated motor cortex (*Nieuwenhuys and Broere, 2017*). A non-neuronal cluster (yellow, *Figure 3e*, *Supplementary file 2*, example core genes: TGFBR2, GMFG, A2M) – which encompassed marker sets for microglia, astrocytes, endothelial cells, pericytes, and markers of superficial adult and fetal cortical layers that are relatively depleted of neurons – was most connected to the M6 gene set highlighting medial temporal and anterior cingulate cortices with notably high non-neuronal content (*Collins et al., 2010*).

These analyses show that the regional patterning of bulk gene expression captures the organization of the human cortex across multiple spatial scales and developmental stages such that (i) the summary expression maps of spatially co-expressed gene sets align with independent in vivo maps of macroscale structure and function from neuroimaging, while (ii) the spatially co-expressed gene sets defining these maps show congruent enrichments for specific adult cortical layers and cell types as well as developmental precursors of these features spanning back to mid-fetal life.

## ASD risk genes follow two different spatial patterns of cortical expression, which capture distinct aspects of cortical organization and differentially predict cortical changes in ASD

The findings above establish that gene co-expression modules in the human cortex capture multiple levels of biological organization ranging from subcellular organelles to cell types, cortical layers, and macroscale patterns of brain structure and function. Given that genetic risks for atypical brain development presumably play out through such levels of biological organization, we hypothesized that disease-associated risk genes would be enriched within WGCNA module gene sets. Testing this hypothesis simultaneously offers a means of further validating our analytic framework, while also potentially advancing understanding of disease biology. To test for disease gene enrichment in WGCNA modules, we compiled lists of genes enriched for deleterious rare variants in ASD (*Ruzzo et al., 2019*; *Satterstrom et al., 2020*), schizophrenia (Scz; *Singh et al., 2020*), severe developmental disorders (DDD; *Deciphering Developmental Disorders Study, 2017*), and epilepsy (*Heyne et al., 2018*; *Supplementary file 2*). We considered rare (as opposed to common) genetic variants to focus on high effect-size genetic associations and avoid ongoing uncertainties regarding the mapping of common variants to genes (*Tam et al., 2019*). We observed that disease-associated gene sets were significantly enriched in several WGCNA modules (*Figure 4a*), with two modules showing enrichments for more than one disease: M15 (ASD, Scz, and DDD) and M12 (ASD and epilepsy). ASD was the only disorder to show a statistically significant enrichment of risk genes within both M12 and M15 (*Figure 4a*), providing an ideal setting to ask if and how this partitioning of ASD risk genes maps onto (i) multiscale brain organization in health and (ii) altered brain organization in ASD.

The eigenmaps and gene set enrichments of M12 vs. M15 implicated two contrasting multiscale motifs in the biology of ASD (*Figure 4b*). ASD risk genes, including SCN2A, SYNGAP1, and SHANK2, resided within the M12 module (*Figure 4c*), which is most highly expressed within a distributed cortical system that is activated during social reasoning tasks ($p_{spin}$<0.01, *Figure 3c and d*, *Figure 4b*). The M12 gene set is also enriched for: genes with peak cortical expression in late-fetal and early postnatal life; marker genes for the fetal subplate and developing excitatory neurons; markers of layer 2 and mature neurons in adult cortex; and synaptic genes involved in neuronal communication (*Figures 3c and d and 4b–e*, *Supplementary file 4*). In contrast, ASD risk genes, including ADNP, KMT5B, and MED13L, resided within the M15 module (*Figure 4c*), which is most highly expressed in primary visual cortex and associated ventral temporal pathways for object recognition/interpretation (*Kravitz et al., 2013*) ($p_{spin}$<0.05, *Figures 3c and d and 4b*, *Supplementary file 4*). The M15 module is also enriched for

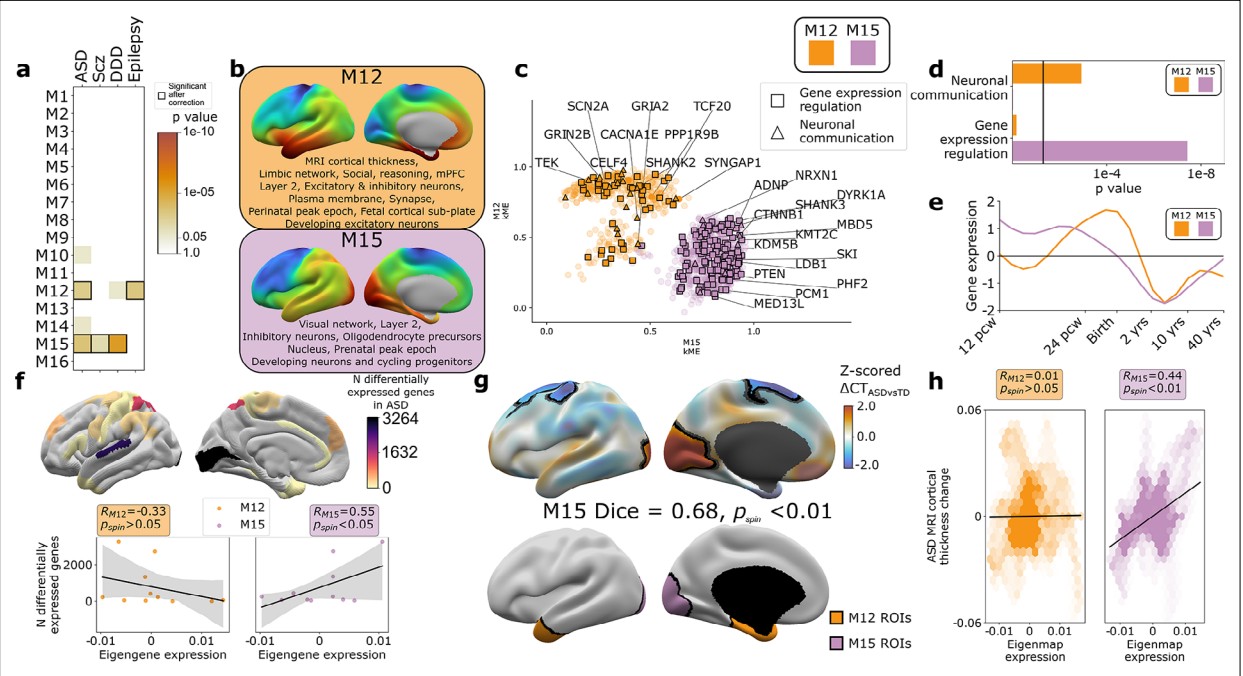

**Figure 4.** Autism spectrum disorder (ASD) risk genes follow two different spatial patterns of cortical gene expression which differentially predict cortical changes in ASD. (**a**) Enrichment of weighted gene co-expression network analysis (WGCNA module gene sets for risk genes associated with atypical brain development through enrichment of rare deleterious variants in studies of ASD, schizophrenia (Scz), severe developmental disorders (DDD, deciphering developmental disorders study), and epilepsy. Cell color intensity indicates pairwise statistical significance (p<0.05)), while outlined matrix cells survived correction for multiple comparisons across modules. (**b**) Summary of multiscale and developmental annotations from *Figure 3c* for M12 and M15: the only two WGCNA modules enriched for risk genes of more than one neurodevelopmental disorder. (**c**) M12 and M15 genes clustered by the strength of their membership to each module. Color encodes module membership. Shape encodes annotations for two GO Biological Process annotations that differ between the module gene sets: neuronal communication and regulation of gene expression. Text denotes specific ASD risk genes. (**d**) Contrasting GO enrichment of M12 and M15 for neuronal communication and regulation of gene expression GO Biological Process annotations. (**e**) M12 and M15 differ in the developmental trajectory of their average cortical expression between early fetal and mid-adult life (*Li et al., 2018*). (**f**) Regional differences in intrinsic expression of the M15 module (but not the M12 module) in adult cortex is correlated with regional variation in the severity of altered cortical gene expression (number of differentially expressed genes) in ASD (*Haney et al., 2020*). (**g**) Statistically significant regional alterations of cortical thickness (CT) in ASD compared to typically developing controls from in vivo neuroimaging (*Di Martino et al., 2017*) (top). Areas of cortical thickening show a statistically significant spatial overlap (Dice overlap = 0.68, p_spin<0.01) with regions of peak intrinsic expression for M15 in adult cortex (bottom). (**h**) M15 eigenmap expression (but not M12 eigenmap) shows significant spatial correlation with relative CT change in ASD.

The online version of this article includes the following figure supplement(s) for figure 4:

**Figure supplement 1.** Additional characterisations of Autism Spectrum Disorder (ASD) risk genes and modules.

genes showing peak cortical expression in early fetal development, marker genes for cycling progenitor cells in the fetal cortex; markers of layer 2, inhibitory neurons and oligodendrocyte precursors in the adult cortex (*Figures 3c and d and 4b–e*, *Supplementary file 4*). The alignment of ASD risk genes with M12 and M15 was reinforced when considering all 135 ASD risk genes: spatial co-expression analyses split these genes into two clear subsets with mean expression maps that most closely resembled M12 and M15 (*Figure 4—figure supplement 1a, b*). Thus, using only spatial patterns of cortical gene expression in adulthood, our analytic framework was able to recover the previous PPI and GO-based partitioning of ASD risk genes into synaptic vs. nuclear chromatin remodeling pathways (*Parikshak et al., 2013*; *Satterstrom et al., 2020*), and then place these pathways into a richer biological context based on the known multiscale associations of M12 and M15 (*Figures 3c and 4a*).

We next sought to address whether regional differences in M12 and M15 expression were related to regional cortical changes observed in ASD. To test this idea, we used two orthogonal indices of cortical change in ASD that capture different levels of biological analysis – the number of differentially expressed genes (DEGs) postmortem (*Haney et al., 2020*), and the magnitude of changes in cortical thickness (CT) as measured by in vivo sMRI (*Di Martino et al., 2017*). Regional DEG counts

were derived from a recent postmortem study of 725 cortical samples from 11 cortical regions in 112 ASD cases and controls (*Haney et al., 2020*), and compared with mean M12 and M15 expression within matching areas of a multimodal MRI cortical parcellation (*Glasser et al., 2016*). The magnitude of regional transcriptomic disruption in ASD was statistically significantly positively correlated with region expression of the M15 module ($r = 0.6$, $p_{spin} < 0.05$), but not the M12 module ($r = -0.3$, $p_{spin} > 0.05$) (*Figure 4f*). This dissociation is notable because M15 (but not M12) is enriched for genes involved in the regulation of gene expression (*Figure 4d*). Thus the enrichment of regulatory ASD risk genes within M15, and the intrinsically high expression of M15 in occipital cortex may explain why the occipital cortex is a hotspot of altered gene expression in ASD.

To compare M12 and M15 expression with regional variation in cortical anatomy changes in ASD, we harnessed the multicenter ABIDE datasets containing brain sMRI scans from 751 participants with idiopathic ASD and 773 controls (*Di Martino et al., 2017*; *Di Martino et al., 2013*). We preprocessed all scans using well-validated tools for harmonized estimation of cortical thickness (CT) (*Fischl, 2012*) from multicenter data ('Materials and methods'), and then modeled CT differences between ASD and control cohorts at 150,000 points (vertices) across the cortex ('Materials and methods'). This procedure revealed two clusters of statistically significant CT change in ASD ('Materials and methods,' *Figure 4g*, upper panel) encompassing visual and parietal cortices (relative cortical thickening vs. controls) as well as superior frontal vertices (relative cortical thinning). The occipital cluster of cortical thickening in ASD showed a statistically significant spatial overlap with the cluster of peak M15 expression (*Figure 4g*, upper panel, 'Materials and methods,' Dice coefficient = 0.7, $p_{spin} < 0.01$), and relative cortical thickness change correlated with the M15 eigenmap (*Figure 4h*). In contrast, M12 expression was not significantly aligned with CT change in ASD (*Figure 4g and h*). Testing these relationships in the opposite direction, that is, asking whether regions of peak M12 and M15 expression are enriched for directional CT change in ASD relative to other cortical regions, recovered the M15-specific association with regional cortical thickening in ASD (*Figure 4—figure supplement 1c*).

Taken together, the above findings reveal that an occipital hotspot of altered gene expression and cortical thickening in ASD overlaps with an occipital hotspot of high expression for a subset of ASD risk genes. These ASD risk genes are spatially co-expressed in a module enriched for several connected layers of biological organization (*Figures 3c and 4b–d*) spanning: nuclear pathways for chromatin modeling and regulation of gene expression; G2/M phase cycling progenitors and excitatory neurons in the mid-fetal cortex; oligodendrocytes and layer 2 cortical neurons in adult cortex; and occipital functional networks involved in visual processing. These multiscale aspects of cortical organization can now be prioritized as potential targets for a subset of genetic risk factors in ASD, and the logic of this analysis in ASD can now be generalized to any disease genes of interest.

## Discussion

We build on the most anatomically comprehensive dataset of human cortex gene expression available to date (*Hawrylycz et al., 2012*), to generate, validate, characterize, apply, and share spatially dense measures of gene expression that capture the topographically continuous nature of the cortical mantle. By representing patterns of human cortical gene expression without the imposition of a priori boundaries (*Burt et al., 2018*; *Hawrylycz et al., 2015*), our library of DEMs allows anatomically unbiased analyses of local gene expression levels as well as the magnitudes and directions of local gene expression change. This core spatial property of DEMs unlocks several methodological and biological advances. First, the unparcellated nature of DEMs allows us to agnostically define cortical zones with extreme transcriptional profiles or unusually rapid transcriptional change, which we show to capture microstructural cortical properties and align with folding and functional specializations at the macroscale (*Figure 2*). By establishing that some of these cortical zones are evident at the time of cortical folding, we lend support to a 'protomap' (*O'Leary, 1989*; *O'Leary et al., 2007*; *Rakic, 1988*; *Rakic et al., 2009*)-like model where the placement of some cortical folds is setup by rapid tangential changes in cyto-laminar composition of the developing cortex (*Ronan et al., 2014*; *Toro and Burnod, 2005*; *Van Essen, 2020*). The DEMs are derived from fully folded adult donors, and therefore some of the measured genetic-folding alignment might also be induced by mechanical distortion of the tissue during folding (*Heuer and Toro, 2019*; *Llinares-Benadero and Borrell, 2019*). However, no data currently exist to conclusively assess the directionality of this gene-folding relationship.

We show that DEMs can recover sharp boundaries in gene expression despite being generated by interpolation algorithms that do not explicitly encode step changes in expression between cortical regions. This property of DEMs will help to target future studies of human cortical patterning (e.g., directing single-cell and spatial omics resources), and we illustrate this utility by applying DEMs to discover two new expression boundaries in the human cortex. Second, we use spatial correlations between DEMs to decompose the complex topography of cortical gene expression into a smaller set of cortex-wide transcriptional programs that capture distinct aspects of cortical biology – at multiple spatial scales and multiple developmental epochs (*Figure 3*). This effort provides an integrative model that links expression signatures of cell types and layers in prenatal life to the large-scale patterning of regional gene expression in the adult cortex, which can in turn, through DEMs, be compared to the full panoply of in vivo brain phenotypes provided by modern neuroimaging. Indeed, future work might find direct links between these module eigenvectors and similar low-frequency eigenvectors of cortical geometry have been used as basis functions to segment the cortex (*Lefèvre et al., 2018*) and explain complex functional activation patterns (*Pang et al., 2023*). Third, we find that some of these cortex-wide expression programs in adulthood are enriched for disease risk genes, which offers a new path to nominating candidate disease mechanisms across different levels of biological organization (*Figure 4*). For example, the M15 module defines a normative spatial pattern of cortical gene co-expression which not only captures a functionally enriched subset of ASD genes (*Satterstrom et al., 2020*), but also shows multiscale enrichments and regionally specific expression patterns that tie together several independently reported aspects of ASD neurobiology. Specifically, M15 newly integrates (i) the concentration of ASD risk genes and dysregulated gene expression in upper-layer excitatory neurons (*Velmeshev et al., 2019*), (ii) the accentuation of altered gene expression and thickness in occipital cortical regions, and (iii) the early emergence among children at heightened genetic risk for ASD of behaviorally relevant changes in cortical structure and function (*Girault et al., 2022*) within occipital systems important for the processing of visual information. Crucially, the strategy applied in our analysis of ASD risk genes can be generalized to risk genes for any brain disorder of interest to place known risk factors for disease into the rich context of multiscale cortical biology.

Finally, the collection of DEMs, annotational gene sets, and statistical tools used in this work is shared as a new resource to accelerate multiscale neuroscience by allowing flexible and spatially unbiased translation between genomic and neuroanatomical spaces. Of note, this resource can easily incorporate any future expansions of brain data in either neuroanatomical or genomic space. We anticipate that it will be particularly valuable to incorporate new data from the nascent, but rapidly expanding fields of high-throughput histology (*Wagstyl et al., 2020*), single-cell omics (*Bakken et al., 2021*), and large-scale imaging-genetics studies (*Smith et al., 2021*). Taken together, MAGICC enables a new integrative capacity in the way we study the brain, and hopefully serves to spark new connections between previously distant datasets, ideas, and researchers.

## Materials and methods
### Creating spatially dense maps of human cortical gene expression (Figure 1a–d)

Cortical surfaces were reconstructed for each AHBA donor MRI using FreeSurfer (*Fischl, 2012*), and coregistered between donors using surface matching of individuals' folding morphology (MSMSulc) (*Robinson et al., 2018*). An average donor cortical mesh was also created for analyses of cortical morphology by averaging the vertex coordinates of volumetrically aligned meshes for the six donors.

Probe-level data measures of gene expression for all samples in the AHBA adult brain microarray dataset were downloaded from https://human.brain-map.org/static/download, providing log2-transformed measures of gene expression for 58,692 probes in each of 3702 brain tissue samples from six donors (*Supplementary file 1*). Within- and across-brain normalization of these probe-level gene expression values was implemented as detailed by the Allen Institute for Brain Science White Paper (*Allen Human Brain Atlas, 2013*). Probes were reannotated using the updated manifest from *Arnatkeviciute et al., 2019*, excluding genes lacking an Entrez, and probe-level expression values were averaged for each gene to yield a single gene * sample expression matrix for each donor. As only two donors had measurements from right hemispheres, samples were filtered by region to retain those originating from the cerebral cortex left hemisphere only. This decision was made given evidence

for potential asymmetries in gene expression within the human cortex (*de Kovel et al., 2018*), and known differences in cortical shape between the hemispheres that complicate the reflection of sample locations from left to right cortical sheets (*Jo et al., 2012*). The above steps resulted in a final set of six donor-level gene * sample matrices from the left cerebral cortex for downstream analyses. These matrices collectively contained scaled expression values for 20,781 genes in each of the 1304 cortical samples.

Native subject MRI coordinates were extracted for every cortical sample in each donor (*Figure 1a*). Nearest mid-surface cortical vertices were identified for each sample, excluding samples further than 20 mm from a cortical coordinate. For cortical vertices with no directly sampled expression, expression values were interpolated from their nearest sampled neighbor vertex on the spherical surface (*Moresi and Mather, 2019*; *Figure 1b*). Sampling density $\rho$ in each subject was calculated as the number of samples per mm², from which average inter-sample distance, *d*, was estimated using the formula: $d = \frac{1}{\sqrt{}}$, giving a mean intersample distance of 17.7 mm ± 1.2 mm. Surface expression maps were smoothed using the Connectome Workbench toolbox (*Glasser et al., 2013*) with a 20 mm full-width at half maximum (FWHM) Gaussian kernel, selected to be consistent with this sampling density (*Figure 1c*). To align subjects' expression, expression values were z-scored by the mean and standard deviation across vertices (given the known criticality of within-subject scaling of AHBA expression values; *Markello et al., 2021*) and then averaged across the six subjects (*Figure 1d*), yielding spatially dense estimates of expression at 29,696 vertices across the left cerebral cortex per gene. Vertex-wise, rather than sample-level, estimation of mean and standard deviation mitigates potential biases introduced by intersubject variability in the regional distribution and density of cortical samples. For Y-linked genes, DEMs were calculated from male donors only. For each of the resulting 20,781 gene-level expression maps, the orientation and magnitude of gene expression change at each vertex (i.e., the gradient) were calculated for folded, inflated, spherical, and flattened mesh representations of the cortical sheet using Connectome Workbench's metric gradient command (*Glasser et al., 2013*).

## Benchmarking DEMs
### Spin tests for comparing two spatial maps
Cortical maps exhibit spatial autocorrelation that can inflate the false-positive rate, for which a number of methods have been proposed (*Alexander-Bloch et al., 2018*; *Burt et al., 2020*; *Vos de Wael et al., 2020*). At higher degrees of spatial smoothness, this high false-positive rate is most effectively mitigated using the spin test (*Alexander-Bloch et al., 2018*; *Markello and Misic, 2021*; *Vos de Wael et al., 2020*). In the following analyses when generating a test statistic comparing two spatial maps, to generate a null distribution, we computed 1000 independent spins of the cortical surface using https://netneurotools.readthedocs.io and applied it to the first map whilst keeping the second map unchanged. The test statistic was then recomputed 1000 times to generate a null distribution for values one might observe by chance if the maps shared no common organizational features. This is referred to throughout as the 'spin test' and the derived p-values as $p_{spin}$.

An additional null dataset was generated to test whether intrinsic geometry of the cortical mesh and its impact on interpolation for benchmarking analyses of DEMs and gradients (*Figure 1—figure supplements 1d and 2d*, *Figure 2—figure supplement 1c*). In these analyses, the original samples were rotated on the spherical surface prior to subsequent interpolation, smoothing, and gradient calculation. Due to computational constraints, the full dataset was recreated only for 10 independent spins. These are referred to as the 'spun + interpolated null.

### Replicability and independence from cortical sampling density (*Figure 1—figure supplement 1*)
We assessed the replicability of DEMs by applying the above steps for DEM generation to non-overlapping donor subsets and comparing DEMs between the resulting sub-atlases. We quantified DEM agreement between sub-atlases at both the gene level (correlation in expression across vertices for each gene, *Figure 1—figure supplement 1c*) and the vertex level (correlation in ranking of genes by their scaled expression values at each vertex, *Figure 1—figure supplement 1d, e*). These sub-atlas comparisons were done between all possible pairs of individuals, donor duos, and donor triplets to give distributions and point estimates of reproducibility for atlases formed of one, two, and three donors. Learning curves were fitted to these data to estimate the projected gene-level and

vertex-level DEM reproducibility of our full six-subject sample atlas (*Figueroa et al., 2012*; *Figure 1—figure supplement 1c*).

To assess the effect of data interpolation in DEM generation, we compared gene-level and vertex-level reproducibility of DEMs against a 'ground truth' estimate of these reproducibility metrics based on uninterpolated expression data. To achieve a strict comparison of gene expression values between different individuals at identical spatial locations, we focused these analyses on the subset of AHBA samples where a sample from one subject was within 3 mm geodesic distance of another. This resulted in 1097 instances (spatial locations) with measures of raw gene expression of one donor and predicted values from the second donor's uninterpolated AHBA expression data and interpolated DEM. We computed gene-level and vertex-level reproducibility of expression using the paired donor data at each of these sample points – for both DEM and uninterpolated AHBA expression values. By comparing DEM reproducibility estimates with those for uninterpolated AHBA expression data, we were able to quantify the combined effect of interpolation and smoothing steps in DEM generation. We used gene-level reproducibility values from DEMs and uninterpolated AHBA expression data to compute a gene-level difference in reproducibility, and we then visualized the distribution of these difference values across genes (*Figure 1—figure supplement 1a*). We used gene–rank correlation to compare vertex-level reproducibility values between DEMs and uninterpolated AHBA expression data (*Figure 1—figure supplement 1b*).

Theoretically, regional gradients of expression change in DEMs could be biased by regional variations in the density of AHBA cortical sampling. To test for this, in each individual subject, we calculated the spatial relationship between the sampling density and mean gene gradient magnitude (*Figure 1—figure supplement 1g*). We additionally tested whether the regional variability of gene rank predictability in the atlas (shown in *Figure 1—figure supplement 1f*) was linked to the sampling density within the atlas.

## Alignment with reference measures of cortical organization (*Figure 1e–g*)

We first determined whether our DEM library was able to differentiate between genes that are known to show cortical expression (CExp) and those without any prior evidence of cortical expression (NCExp) – motivated by the strong expectation that NCExp genes should lack a consistent spatial gradient in expression. For this test, we defined a set of 16,573 CExp genes by concatenating the genes coding for proteins found in the 'cortex' tissue class of the human protein atlas (*Sjöstedt et al., 2020*) genes identified as markers for cortical layers or cortical cells (see below; *Darmanis et al., 2015*; *Habib et al., 2017*; *He et al., 2017*; *Hodge et al., 2019*; *Lake et al., 2018*; *Lake et al., 2016*; *Li et al., 2018*; *Maynard et al., 2021*; *Ruzicka et al., 2021*; *Velmeshev et al., 2019*; *Zhang et al., 2016*). The remaining 4208 genes in our DEM library were classified as NCExp. Fisher's exact test was used to assess whether genes with lower gene reproducibility ($r < 0.5$) were enriched for NCExp genes. We projected vertex-level reproducibility values for CExp and NCExp genes onto the cortical surface for visual comparison and also computed the mean cross-vertex reproducibility for each of these maps (*Figure 1—figure supplement 1f*).

We next compiled data from independent studies for a range of macroscale and microscale cortical features that would be expected to align with specific DEM maps, and asked whether the spatial patterns of cortical gene expression from DEMs showed the expected alignment with these independent data. These independent comparison studies were selected to span diverse measurement methods and data modalities representing a range of spatial scales.

We first sought to establish whether local changes in DEMs, that is, the gradient maps of gene expression, could be used to validate existing areal border genes and identify novel candidates. Using a parcellation of the cortex based on multimodal structural and functional neuroimaging (*Glasser et al., 2016*), we identified the vertices along the boundary between a pair of regions (e.g., V1 and V2). The mean DEM gradient at these vertices was quantified for each gene, enabling us to rank genes by their exhibited border-like features at this cortical location. We then assessed the ranking of known lists of areal marker genes for a given border against a randomly sampled null distribution. To validate known areal marker genes derived from previous ISH studies, we took examples from the human visual cortex (*Zeng et al., 2012*), macaque visual cortex, and macaque frontal regions 44 and 45 (*Chen et al., 2022*). To test the capacity of our resource to identify novel putative areal border genes, we calculated average gradients of all genes across the boundary between mesial temporal

**Table 1.** Statistical tests used to compare spatial maps and gene sets derived from the Allen Human Brain Atlas with independent multiscale neuroscientific resources.

| Input data | Test statistic | Significance test |
|---|---|---|
| Comparison of two cortical maps, e.g., *Figure 1e* | Pearson's R (e.g., *Figure 1e and f*), Spearman $R_{rank}$ (*Figure 3*), delta Z for binary and continuous comparison (*Figures 1e and 3c and d*), Dice score for two binary maps (*Figures 2d and 4g*), skew in distribution of angles (*Figures 2f and c and 3cFigures 2f & g and 3c*). Counts for peak expression locations overlapping ROIs (*Figure 2—figure supplement 1h*). | Spin test: generate null distribution for test statistic by independently spinning spherical projections of spatial maps and recalculating test statistic on spun maps (*Alexander-Bloch et al., 2018*). |
| Intrasubject alignment of multimodal maps | Pearson's R (e.g., *Figure 1—figure supplement 1g*). | Simple permutation-based intermodal correspondence (SPICE) test (*Weinstein et al., 2021*). |
| Comparison of gene–gene connectivity matrix, e.g., protein–protein interaction vs. spatial correlation, gene–gene spatial correlation vs. developmental trajectory correlation | If continuous, threshold matrix at 95%. Fisher's exact test for significant edge-level overlap. | Fisher's exact test p-values corrected for multiple comparisons using the Holm–Sidak step down procedure (*Holm, 1979*). |
| Overlap of two gene lists, e.g., *Figure 3e* | Fisher's exact test. | Fisher's exact p-value corrected for multiple comparisons. |
| Cortical thickness changes in pathology (in AD, *Figure 1e*, in ASD, *Figure 4g*) | Linear model: Vertex cortical thickness ~ Age + sex + group + mean cortical thickness. | Cluster-wise correction. Calculate maximum size of significant clusters on 1000 randomly permuted cohorts, using the 95th centile size as a threshold on the test cohort (*Hagler et al., 2006*). |
| Intramodular trajectory correlation | Pairwise intramodular median rank correlation. | Randomly sampled gene sets of comparable size. |
| Protein–protein interaction | Intramodular connectivity. | Random resampling of gene sets with decile-matching for degree. |

AD, Alzheimer's disease; ASD, autism spectrum disorder; ROI, region of interest.

parahippocampal gyrus (perirhinal ectorhinal cortex, PeEc) and the fusiform gyrus (area TF) for which there is openly available ISH data (https://human.brain-map.org/ish/search). Limiting analyses to those genes for which ISH was available, the two genes exhibiting the largest gradient in either direction (four in total) were selected. The ISH was visually inspected for the presence of area-like features in gene expression. For quantitative support, the cortex in each ISH image was manually segmented over the area of interest. The pixel-wise transverse distance along the cortical segmentation from left to right was calculated and subdivided into 200 equally spaced columns, spanning from pial to white matter surface. Staining intensity was averaged across each column. For each column, we computed the *t*-statistic between columns to the right and left, and identified the column with the largest *t*-statistic as the location of the putative interareal boundary.

We used the spun + interpolated null to test whether peaks in gene gradient could be driven primarily by local folding morphology impacting non-uniform interpolation. We quantified the average gradient for all genes along the V1-V2 border in the atlas, as well as for 10 iterations of the atlas where the samples were spun prior to interpolation. We computed the median gradient magnitude for the 20 top-ranked genes for each (*Figure 1—figure supplement 2d*).

We benchmarked DEMs against regional differences in cellular measures of cortical organization from single-nucleus RNA-sequencing studies (snRNAseq). Specifically, we correlated regional differences in the estimated proportion of 16 neuronal subtypes across six cortical regions (*Lake et al., 2016*) with regional DEM estimates for the mean expression of provided markers for these cell types (*Lake et al., 2016*). The test statistic was tested against a null distribution generated through spinning and resampling the cell marker DEM estimates (*Table 1*). Given the observed correspondence between regional cellular proportions and regional expression of cell marker sets, we used more recently generated reference cell markers from the Allen Institute for Brain Sciences (*Bakken et al., 2021*; *Hodge et al., 2019*; *Tasic et al., 2016*) to generate DEMs for 11 of 14 major cell subclasses in the mammalian cortex (6 neuronal types shown in *Figure 1h*, all 11 used for TD peak enrichment

analysis; *Figure 2—figure supplement 1h*). Three markers were excluded due to absence in the original dataset or low gene predictability (*r* < 0.2, *Figure 1—figure supplement 1c*).

We benchmarked DEMs against orthogonal spatially dense measures of cortical through the following comparisons: (i) layer IV thickness values from the 3D BigBrain atlas of cortical layers (*Wagstyl et al., 2020*) vs. the average DEM for later IV marker genes (*He et al., 2017*; *Maynard et al., 2021*; *Supplementary file 2*); (ii) motor-associated areas of the cortex from multimodal in vivo MRI (*Glasser et al., 2016*) vs. the average DEM for two marker genes (ASGR2, CSN1S1) of Betz cells, which are giant pyramidal neurons that output from layer V of the human motor cortex (*Bakken et al., 2021*); (iii) an in vivo neuroimaging map of the T1/T2 ratio measuring of intracortical myelination (*Glasser and Van Essen, 2011*) vs. the DEM for Myelin Basic Protein; and (iv) regional cortical thinning from in vivo sMRI data in AD patients with the APOE E4 (OASIS-3 dataset [*LaMontagne et al., 2019*], see 'MRI data processing') vs. the APOE4 DEM. For all four of these comparisons, alignment between maps was quantified and test for statistical significance using a strict spin-based spatial permutation method that controls for spatial autocorrelation in cortical data (*Alexander-Bloch et al., 2018*) methods on statistical testing of pairwise cortical maps can be found in *Table 1*.

## Characterizing the topography of DEMs
### TD and PCA (*Figure 2a–c*)
TD of each cortical vertex was calculated as the mean of the absolute DEM value for all genes (*Figure 2a*). Statistically significant peaks in TD, driven by convergence of extreme values across multiple genes, were identified as follows. The DEM for each gene was independently spun and TD was recalculated at each vertex over 1000 sets of gene-level DEM permutations (*Alexander-Bloch et al., 2018*). The maximum vertex TD value for each permuted TD map was recorded and the 95th percentile value across the 1000 permutations was taken as a threshold value. This threshold represents the maximum TD one would expect in the absence of concentrated colocalisations of extreme expression signatures, and areas above this threshold were annotated as TD peaks. To disambiguate TD peaks that are spatially coalescent but potentially driven by extreme values of heterogeneous gene sets within different regions, we concatenated all suprathreshold TD vertices into a single vertex * gene matrix and vertices in this matrix were clustered based on their expression signatures.

Intervertex correlation of gene rankings was calculated and the matrix was clustered using a Gaussian mixture model. Bayesian information criterion was used to identify the optimum number of clusters (k = 6) from a range of 2–18. Automated labels to localize TD peaks were generated based on their intersection with a reference multimodal neuroimaging parcellation of the human cortex (*Glasser et al., 2016*). Each TD was given the label of the multimodal parcel that showed the greatest overlap (*Figure 2b*).

The TD map was assessed for reproducibility through three approaches. First, the six-subject cohort was subdivided into pairs of triplets, for which there are 10 unique combinations. For each combination, independent TD maps were computed for each triplet and compared between triplets (*Figure 2—figure supplement 1a*). Second, for the full six-subject cohort genes were grouped into deciles according to the reproducibility of their spatial patterns in independent subcohorts (*Figure 1—figure supplement 1c*). For each decile of genes, a TD map was computed and compared to the TD map from the remaining 90% of genes (*Figure 2—figure supplement 1b*). Third, to assess whether the covariance in spatial patterning across genes could be a result of mesh-associated structure introduced through interpolation and smoothing, TD maps were recomputed for the spun + interpolated null datasets and compared to the original TD map (*Figure 2—figure supplement 1c*).

The cortical regions defined by TD peaks were annotated according to their spatial overlap with the 24 cortical cell marker expression DEMs used in *Figure 1g and h* (*Bakken et al., 2021*; *Hodge et al., 2019*; *Lake et al., 2016*). To establish that cell maps were aligned with TD peaks, we first tested whether the vertex with the highest DEM value for each cell map overlapped with a TD peak and compared the number of overlapping cells to a null distribution created through spinning the TD peaks independently 1000 times. We then identified the cell types whose expression most closely aligned with each TD peak, comparing mean TD expression with a null distribution generated through spinning the peaks 1000 times (*Figure 2—figure supplement 1h*). TD peaks were also annotated for their functional activations using the meta-analytic Neurosynth database (see 'Map-based annotations').

Gene sets characterizing TD peaks were identified as follows. At the vertex with the highest TD value within a peak region, the 95th centile TD value across genes was selected as a threshold. Genes with z-scored expression values above this threshold or below its inverse were selected, allowing TD peaks to have asymmetric length gene lists for high- and low-expressed genes (*Supplementary file 3*). These TD gene lists were submitted to a Gene Ontology (GO) enrichment analysis pipeline (see 'Gene set-based annotations').

To contextualize the newly described TD peaks using previously reported principal components (PCs) of human cortical gene expression, we computed the first five PC of gene expression in our full DEM library. The percentage of variance explained by each PC was calculated and compared to a null threshold derived through fitting PCs to a permuted null given by 1000 random spatial rotations of gene-level DEMs (*Figure 2—figure supplement 1d*). Taking the gene-level loadings from the first three PCs (*Figure 2—figure supplement 1e*), each vertex could be positioned in a 3D PC space based on its expression signature and also be colored based on its membership of a TD peak, thereby visualizing the position of TD peaks relative to the dominant spatial gradients of transcriptomic variation across the cortex (*Figure 2c*).

The assignment of TD regions as 'peaks' implies a rapid emergence of the TD signature surrounding the peak boundaries, which we formally assessed by cortex-wide analysis of local tangential changes in gene expression (see 'Local gradient analysis'), and a spatially fine-grained comparisons of the physical vs. transcriptional distance between cortical regions. In the latter of these two analytic approaches, a rapid 'border-like' onset of TD features would appear as (i) TD regions showing a greater transcriptional distance from other cortical regions than would be expected from their physical distance from other cortical regions, and (ii) this disparity emerging sharply surrounding the peak. To achieve this test, we first quantified the geodesic physical distance and Euclidean transcriptomic distance between pairs of vertices. For computational tractability, we limited this analysis to a subsample of vertices, choosing central vertices from ROIs in a parcellation with 500 approximately evenly sized parcels (*Schaefer et al., 2018*). We fit a linear generalized additive model to the data – predicting transcriptomic distance from geodesic distance – and calculated the residuals for each inter-vertex edge (*Figure 2—figure supplement 1f*). For each sampled vertex, we averaged these residuals and mapped them back to the surface to visualize cortical areas that were transcriptomically more distinctive than their physical distance to other areas would predict (*Figure 2—figure supplement 1g*).

## Relating adult TD peaks to fetal gene expression (*Figure 2—figure supplement 1k*)

We sought to establish whether the regional expression signatures characterizing TD peaks were present early in fetal development. This goal required measures of gene expression from multiple regions across the fetal cortical sheet, which are provided by the Allen Institute from Brain Sciences fetal laser micro-dissection microarray dataset (*Miller et al., 2014*). In each sample's fetal brain, this dataset represents approximately 25 cortical brain regions tangentially, and radially 7 transient fetal layers/compartments: subpial granular zone (SG), marginal zone (MZ), outer and inner cortical plate (grouped together as CP), subplate zone (SP), intermediate zone (IZ), outer and inner subventricular zone (grouped together as SZ), and ventricular zone (VZ).

Probe-level data measures of gene expression for the two PCW21 donors in the AHBA fetal LMD microarray dataset were downloaded from https://www.brainspan.org/static/download.html, providing log2-transformed measures of gene expression for 58,692 probes in each of the 536 tissue samples across both donors (*Supplementary file 1*). Preprocessing and normalization of these probe-level gene expression values were implemented as detailed by the Allen Institute for Brain Science White Paper (https://help.brain-map.org/download/attachments/3506181/Prenatal_LMD_Microarray.pdf). Probe-level expression values were averaged for each gene to yield a single gene * sample expression matrix for each donor, which was filtered to include only cortical samples. Gene expression values were scaled across samples within each donor, and scaled gene expression values were compiled for the set of 235 cortical regions that was common to both donor datasets. We averaged scaled regional gene expression values between donors per gene and filtered for genes in the fetal LDM dataset that were also represented in the adult DEM dataset, yielding a single final 20,476 * 235 gene-by-sample matrix of expression values for the human cortex at 21 PCW. Each TD peak region was then paired with the closest matching cortical label within the fetal regions. This matrix was then

used to test whether each TD expression signature discovered in the adult DEM dataset (*Figure 2*, *Supplementary file 3*) was already present in similar cortical regions at 21 PCW.

The analysis of fetal regional patterning of TD peak gene sets was carried out as follows (*Figure 2—figure supplement 1k*). For a given TD peak, the significantly enriched genes for that peak (see above for definition of these gene sets) were identified in the fetal dataset and averaged at each fetal sample – capturing how highly expressed the TD signature was in each fetal sample. Next, we identified all samples in the fetal expression dataset that originated from regions underlying the TD peak and defined these as the 'fetal target region set' for that TD region (i.e., occipital samples in the fetal brain were the fetal target region set for analysis of gene enriched in the adult occipital TD region). We ranked all fetal samples by their mean expression of the TD marker set and normalized these ranks to between 0 (TD markers most highly expressed) and 1 (TD markers most lowly expressed). Normalization was done to adjust for varying numbers of areas recorded per compartment. This ranking enabled us to compute the median rank of the fetal target region set and test whether this was significantly lower compared to a null distribution of ranks from random reassignment of the fetal target region set labels across all fetal samples. Within this analytic framework, a statistically significant test means that the adult TD signature is significantly localized to homologous cortical regions at 21 PCW fetal life (*Figure 2—figure supplement 1k* ). We repeated this procedure for each adult TD.

## Local gradient analysis (*Figure 2e–g*)

Spatially DEMs enabled the calculation of a vector describing the first spatial derivative, that is, the local gradient, of each gene's expression at each vertex. These vectors describe both the orientation and the magnitude of gene expression change.

Averaging these gene-level magnitude estimates across genes provided a vertex-level summary map of the magnitude of local expression changes in our full DEM library (*Figure 2e*). Regions with a significantly high average expression gradient were identified using a similar spatial permutation procedure as described for the identification of TD peaks. Briefly, the DEM gradient map for each gene was independently spun and an average expression gradient magnitude was recalculated at each vertex over 1000 sets of these spatial permutations (*Alexander-Bloch et al., 2018*). For each permutation, we recorded the maximum vertex-level average expression gradient value, and the 95th percentile value of these maximums across the 1000 permutations was taken as a threshold value. Vertices with observed average expression gradient values above this threshold represented cortical regions of significantly rapid transcriptional change (*Figure 2—figure supplement 1j*).

The principal orientation of gene expression change at each vertex was calculated considering the vectors describing gene expression gradients, thereby providing a single summary of local gene expression gradients that considers both direction and magnitude. PCA of gene gradient vectors was used to calculate the primary orientation of gene expression change at each vertex (*Figure 2e*) and the percentage of orientation variance accounted for by this PC (*Figure 2e*, *Figure 2—figure supplement 1i*). Gene-level PC weights for each vertex were stored for subsequent analyses, including alignment with folds and functional ROIs (*Figure 2f and g*, see 'Annotational analyses').

The rich DEM expression gradient information described above was applied in three downstream analyses. First, we used these resources to detail the emergence of TD expression signatures within the cortical sheet, focusing on all vertices that had been identified to show a significantly elevated mean expression gradient. Specifically, we ranked genes at these vertices by their loadings onto the first PC of gene expression gradients at each vertex and correlated these rankings with the rankings of genes by the expression at each TD peak vertex. This vertex-level correlation score, which quantifies how closely the gene expression gradient at a given vertex resembles that expression signature of a given TD peak, was regenerated for each of the six TD peaks (colors, *Figure 2—figure supplement 1j*). In each of these six maps, we were also able to plot the principal orientations of expression change at the vertex level (red lines, *Figure 2—figure supplement 1i*) to ask whether gradients of expression change for a given TD signature were spatially oriented towards the TD in question.

Second, we used the principal orientation of expression change at each vertex to assess whether local transcriptomic gradients were aligned with the orientation of cortical folding patterns. Orientation of cortical folds was calculated using sulcal depth and cortical curvature (*Xia et al., 2018*). Gradient vectors for sulcal depth describe the primary orientation of cortical folds on the walls of sulci, while gradient vectors of cortical curvature better describe the orientation at sulcal fundi and

gyral crowns. These two gradient vector fields were combined and smoothed with a 10 mm FWHM Gaussian kernel to propagate the vector field into plateaus, for example, at large gyral crowns where neither sulcal depth nor curvature exhibit reliable gradients. The folding orientation vectors were calculated with reference to a 2D flattened cortical representation for statistical comparison with the gradient vectors derived from gene expression maps (*Figure 2f*). At each vertex, the minimum angle was calculated between the folding orientation vector and gene expression gradient vector. Aligned vector maps exhibit positive skew, with angles tending toward zero. Therefore, the skewness of the distribution of angles across all vertices was calculated, and to test for significance, folding and expression vector maps were spun relative to one another 1000 times, generating a null distribution of skewness values against which the test statistic was compared (*Table 1*). A similar analysis was applied to test the association between module eigenmap gradient vectors and cortical folding (see 'Weighted gene co-expression network analysis (WGCNA)').

Third, we sought to quantify the alignment between cortical expression gradients and cortical areas as defined by multimodal imaging. Orientation of each MRI multimodal parcel ROI from *Glasser et al., 2016* was calculated taking the coordinates for all vertices within a given ROI. PCA of coordinates was used to identify the short and long axis of the ROI object. The vector describing the short axis was taken for comparison with mean of expression gradient vectors for vertices in the same ROI. For each ROI, the minimum angle was calculated and the skewness of the angles across all ROIs was calculated and compared to a null distribution created through spinning maps independently 1000 times, recalculating angles and their skewness (*Figure 2g*).

## Weighted gene co-expression network analysis (WGCNA) (*Figure 3a–c*)

Genes were clustered into modules for further analysis using WGCNA (*Langfelder and Horvath, 2008*). Briefly, gene–gene cortical spatial correlations were calculated across all vertices to generate a single square 20,781 * 20,781 signed co-expression matrix. This co-expression matrix underwent 'soft-thresholding,' raising the values to a soft power of 6, chosen as the smallest power where the resultant network satisfied the scale-free topology model fit of $r^2 > 0.8$ (*Zhang and Horvath, 2005*). Next, a similarity matrix was created through calculating pairwise topological overlap, assessing the extent to which genes share neighbors in the network (*Yip and Horvath, 2007*). The inverse of the TOM was then clustered using average linkage hierarchical clustering, with a minimum cluster size of 30 genes. The eigengene for each module is the first PC of gene expression across vertices and provides a single measure of module expression at each vertex (hence, 'eigenmap'). As per past implementation of WGCNA, pairs of modules with eigengene correlations above 0.9 were merged. These procedures defined a total of 23 gene co-expression modules ranging in size from 77 to 3725 genes, and a single set of unconnected genes (gray module 265 genes). We filtered the gray module from further analysis, as well as all six other modules that were also statistically significantly enriched for NCExp genes (*Supplementary file 4*, Fisher's test, all p<0.0001), leaving a total of 16 modules for downstream analysis (*Supplementary file 4*). To assess the extent to which eigenmaps captured highly reproducible features of cortical organization, for each decile of genes, DEMs were correlated with their module eignmaps recomputed from the remaining 90% of genes (*Figure 3—figure supplement 1a*).

Each WGCNA module could be visualized as a cortical eigenmap, and eigenmap gradient – on the TD terrain, or inflated cortical (*Figure 3a*). The eigenmap gradient for each module provides a vertex-level measure for the magnitude of change in module expression at each vertex, as well as a vertex-level orientation of module expression change – calculated as described in 'Local gradient analysis'. These anatomical representations of each WGCNA module are amenable to spatial comparison with any other cortical map through spatial permutations (*Alexander-Bloch et al., 2018*; see 'Annotational analyses'). Each WGCNA module is also defined as a gene set, which is amenable to standard gene set-based enrichment analysis (see 'Annotational analyses'). WGCNA modules can each also be represented as a ranked list of all genes – based on gene-level kME scores for each module, which are the cross-vertex correlation between a gene's DEM map and a module's eigenmap.

## Multiscale annotation of WGCNA modules (Figure 3c and d)

We used multiple open neuroimaging and genomic datasets to systematically sample diverse levels of cortical organization and achieve a multiscale annotation of WGCNA modules. All gene sets used in enrichment analysis are detailed in *Supplementary file 2*.

## Map-based annotations

### MRI-derived maps of cortical function

Functional annotations of the cortex were carried out using two independent functional MRI (fMRI) resources – one based on state fMRI (rs-FMRI) (*Yeo et al., 2011*) and one using task-based fMRI (*Rubin et al., 2017*; *Yarkoni et al., 2011*). Resting-state functional connectivity networks were taken from *Yeo et al., 2011*, which divides the cortex into seven coherent functional networks through surface-based clustering of rs-fMRI as visual, somatomotor, dorsal attention, ventral attention, frontoparietal control, limbic, and default networks. We used spin-based spatial permutation testing to test for networks in which WGCNA eigenmap expression was significantly elevated (*Figure 3c*, see *Table 1*).

For task fMRI-driven functional annotation of the cortex, we drew on meta-analytic maps of cortical activation from *Neurosynth* (*Rubin et al., 2017*; *Yarkoni et al., 2011*). Briefly, over 11,000 functional neuroimaging studies were text-mined for papers containing specific terms and associated activation coordinates (*Yarkoni et al., 2011*). Secondary analyses generated activation maps for 30 topics spanning a range of cognitive domains (*Rubin et al., 2017*). Topic activation maps were intersected with cortical surface meshes and thresholded to identify vertices with an activation value above 0. Example topics included 'motor, cortex, hand' and 'social, reasoning, medial prefrontal cortex' (*Figure 3d*). Topics were excluded if intersected cortical maps indicated activation in fewer than 1% of cortical vertices. Topic maps were used to annotate TD peaks (*Figure 2d*), identifying for each ROI the two topics with the highest Dice overlap. Topic maps also served as an independent validation of selected WGCNA eigenmaps (*Figure 2d*, *Table 1*). Topic maps from *Neurosynth* were also used to provide an orthogonal validation of observed resting-state network enrichments from Yeo et al (*Figure 3c*) for M2 and M12: mean eigenmap expression for module M2 and M12 was calculated for *Neurosynth* topic maps and assessed for statistical significance using spin-based permutations (*Figure 3d*, *Table 1*).

### MRI-derived maps of cortical structure

Cortical thickness and T1/T2 'myelin' maps were taken from the Human Connectome Project average (*Glasser et al., 2016*). Spatial correlations were calculated across all vertices with each WGCNA module eigenmap and assessed for statistical significance using spin-based permutations (*Figure 3c*, see *Table 1*).

### Orientation of cortical folds

We used the orientation of expression change at each vertex to assess whether local eigenmap gradients were aligned with the orientation of cortical folding patterns, mirroring the analysis described above (*Figure 3—figure supplement 1b*, see 'Local gradient analysis').

### Inter-eigenmap correlations

We tested the pairwise spatial correlation between pairs of module eigenmaps. Statistical significance was assessed using a null distribution of correlation matrices through independently spinning eigenmaps and recalculating correlations, and correcting for multiple comparisons (*Figure 3—figure supplement 1c*, see *Table 1*).

## Gene set-based annotations

### GO enrichment

GO enrichment analysis (see *Table 1*) were carried out on gene sets of interest, testing for enrichment of Biological Processes and Cellular Compartment, using the GOATOOLS Python package (*Klopfenstein et al., 2018*). Where multiple gene lists were assessed simultaneously (e.g., for TD peak gene lists or WGCNA gene sets), correction for multiple comparisons was carried out by dividing the p<0.05 threshold for statistical significance by the number of tests (i.e., for 16 module p<0.05/16). To facilitate summary descriptions of multiple significant GO terms, terms were hierarchically clustered based on semantic similarity (*Resnik, 1995*) and representative terms were selected based on biological specificity (i.e., depth within the Gene Ontology tree) and magnitude of the enrichment statistic (*Figure 3d*, *Supplementary file 2*).

### Layer marker gene sets and ISH validation

We sought to assess the extent to which convergent spatial patterns of gene expression indicate convergent laminar and cellular features. Marker genes for each cortical layer were defined as the union of layer-specific marker genes from two comprehensive transcriptomic studies of layer-dependent gene expression sampling prefrontal cortical regions (*He et al., 2017*; *Maynard et al., 2021*). He et al. took human cortical samples from the prefrontal cortex, corresponding to areas BA 9, 10, and 46. Samples were sectioned into cortical depths and underwent RNAseq to identify 4131 genes exhibiting layer-dependent expression. Maynard et al. took samples from the dorsolateral prefrontal cortex and carried out spatial snRNAseq to identify 3785 genes enriched in specific cortical layers. These independent resources were combined for laminar enrichment analyses (i.e., we took each layer's marker genes to be the union of layer genes defined in Maynard et al. and He et al.). WGCNA module genes were tested for laminar enrichment using Fisher's exact test, correcting for multiple comparisons (*Figure 3c*, see *Table 1*). Independent validation of laminar associations of candidate genes identified through the above marker lists was carried out using ISH data from the Allen Institute (*Zeng et al., 2012*). For selected modules, we identified the highest kME genes represented within the ISH dataset. For each of these genes, the highest quality sections were downloaded, and the cortical ribbon was manually segmented. Equivolumetric estimates of cortical depth were generated and profiles of depth-dependent staining intensity were generated (*Huber et al., 2021*). Accompanying approximate cytoarchitectonic layer thickness estimations were derived from BigBrain and used to describe the laminar location of ISH peaks (*Wagstyl et al., 2020*; *Figure 3d*).

### Adult cortical cell-type marker gene sets

Cell marker gene sets were compiled from multiple snRNAseq datasets, sampling a wide variety of cortical areas covering occipital, temporal, frontal, cingulate, and parietal lobes (*Darmanis et al., 2015*; *Habib et al., 2017*; *Hodge et al., 2019*; *Lake et al., 2018*; *Lake et al., 2016*; *Li et al., 2018*; *Ruzicka et al., 2021*; *Velmeshev et al., 2019*; *Zhang et al., 2016*). To integrate across differing subcategories, cell subtype marker lists were grouped into the following cell classes according to their designated names: excitatory neurons, inhibitory neurons, oligodendrocytes, astrocyte, oligodendrocyte precursor cells, microglia, and endothelial cells. Marker lists for each of these cell classes represented the union of all subtypes assigned to the category. Cells not fitting into these categorizations were excluded. WGCNA module genes were tested for cell class marker enrichment using Fisher's exact test, correcting for multiple comparisons (*Figure 3c*, see *Table 1*).

### Fetal cortical cell-type marker gene sets

Fetal cell marker gene lists were taken from *Polioudakis et al., 2019*. WGCNA module genes were tested for cell class marker enrichment using Fisher's exact test, correcting for multiple comparisons (*Figure 3c*, see *Table 1*).

### Compartments and SynGO

Cellular compartment gene lists were taken from the COMPARTMENTS database (*Binder et al., 2014*), which identifies subcellular localisation of marker genes based on integrated information from the Human Protein Atlas, literature mining, and GO annotations. Examples of cellular compartments include nucleus, plasma membrane, and cytosol. An additional compartment list for neuronal synapse was generated by collapsing all genes in the manually curated SynGO dataset (*Koopmans et al., 2019*). WGCNA module genes were tested for cell compartment gene set enrichment using Fisher's exact test, correcting for multiple comparisons (*Figure 3c*, see *Table 1*).

### PPI network

PPIs were derived from the STRING database (*Szklarczyk et al., 2019*). Physical direct and indirect PPIs were considered. We tested for enrichment of PPIs for proteins coded by genes within WGCNA modules. The median number of intramodular connections was compared to a null distribution of median modular connectivity derived from 10,000 randomly resampled modules with the same number of genes. Gene resampling was restricted within deciles defined by the degree of protein–protein connectivity.

### Developmental peak epoch

Peak developmental epochs for genes were extracted from *Werling et al., 2020*. Briefly, bulk transcriptomic expression values were measured from DLPFC samples across development (6 PCW to 20 y), fitting developmental trajectories to each gene. Genes were categorized according to developmental epoch in which their expression peaked. For descriptive purposes, epochs were renamed as (1) 'early fetal' ('fetal,' 8–24 PCW), (2) late fetal transition ('perinatal,' 24 PCW – 6 mo postnatal), and (3) 'postnatal' (>6 mo). Genes associated with WGCNA modules were tested for enrichment correcting for multiple comparisons across 16 modules.

### Developmental trajectories

Gene-specific developmental trajectories were generated for the cortical samples from *Li et al., 2018*. Briefly, in this study bulk transcriptomic expression values were measured from brain tissue samples taken from individuals aged between 5 PCW and 64 years old. In our analysis, samples were filtered for cortical ROIs and restricted to post 10 PCW due to lack of samples before this time point. Ages were log transformed and Generalized Additive Models were fit to each gene to generate an estimated developmental trajectory. To compute trajectory correlations between genes, we first resampled expression trajectories at 20 equally spaced time points (in log time), and then z-normalized these values per gene (using the mean and standard deviation of each trajectory). We then calculated expression trajectory Pearson correlations between each pair of genes in this dataset and used these to determine whether the spatially co-expressed genes defining each WGCNA module also showed significant temporal co-expression. To achieve this test, we calculated the median temporal co-expression (correlation in expression trajectories) for each WGCNA module gene set and compared this to null median co-expression values for 1000 randomly resampled gene sets matching module size. The mean trajectories of genes in each module were calculated to visualize the developmental expression pattern of each module (*Figure 3—figure supplement 1d*).

### Fetal compartmental analysis

We used the 21 PCW fetal microarray data processed for analysis of TD peaks (see 'Relating adult TD peaks to fetal gene expression,' *Figure 2—figure supplement 1k*; *Miller et al., 2014*) to generated marker gene sets for each of the seven transient fetal cortical compartments: subpial granular zone (SG), marginal zone (MZ), outer and inner cortical plate (grouped together as CP), subplate zone (SP), intermediate zone (IZ), outer and inner subventricular zone (grouped together as SZ), and ventricular zone (VZ). We collapsed 21 PCW cortical expression data into compartments by averaging expression values across cortical regions for each compartment because compartment differences are known to explain the bulk of variation in cortical expression within this dataset (24% [*Miller et al., 2014*]). The top 5% expressed genes for each of the seven fetal compartments was taken as the compartment marker set and used for enrichment analysis of WGCNA modules with Fisher's exact test, correcting for multiple comparisons (see *Table 1*, *Figure 3c*).

### Reproducibility of genes driving enrichment analyses

We calculated gene-level spatial reproducibilities for the union of all genes contributing to significant neurobiological enrichments of WGCNA modules. This was compared to a null distribution, randomly resampling the same number of genes from all those considered in the enrichment analyses.

## Combining gene set-based annotations of the cortical sheet (Figure 3e , Figure 2—figure supplement 1d)

Our observation that many WGCNA modules showed statistically significant enrichment for diverse gene sets that could span different spatial scales (e.g., layers and organelles) or temporal epochs (e.g., fetal and adult cortical features) (*Figure 3c*) suggested a potential sharing of marker gene across these diverse sets. To test this idea and characterize potential biological themes reflected by these shared marker genes, we carried out pairwise enrichment analyses between all annotational gene lists (*Figure 3e*). Gene lists used for enrichment analysis of WGCNA modules for cortical layers, adult cells, cellular compartments, fetal cells, developmental peak epochs, and fetal compartments were taken for further analysis. A genelist–genelist pairwise enrichment matrix was generated. p-Values above 0.1 were set to 1, to limit their contribution, and p-values were converted to -log10(p). To

remove isolated gene lists, all lists were ranked by their degree (edges defined as p<0.05) and the bottom 10% were excluded from further analysis. The matrix, excluding WGCNA modules, underwent Louvain clustering (*Blondel et al., 2008*), grouping together gene lists with similar properties. Clusters were assigned descriptive names according to their salient common features (e.g., non-neuronal, mature neuron, mitotic, myelin, fetal GE) (*Figure 3—figure supplement 1e*). For visualization, the full matrix underwent UMAP embedding (*McInnes et al., 2018*), a nonlinear dimensionality reduction technique assigning 2D coordinates to each gene list (*Figure 3e*), coloring gene lists by their assigned cluster along with the top 20% of edges.

## Disease enrichment and ASD-based analysis of WGCNA modules

The proposed analyses above link regionally patterned cortical gene expression with macroscale imaging maps of structure and function, and microscale gene sets exhibiting laminar, cellular, subcellular, and developmental transcriptomic specificity. We sought to assess whether WGCNA module gene lists capturing shared spatial and temporal features were also enriched for genes implicated in atypical brain development. We included genes identified in exome sequencing studies in neurodevelopmental disorders: ASD (*Ruzzo et al., 2019*; *Satterstrom et al., 2020*) , schizophrenia (SCZ; *Singh et al., 2020*), severe developmental disorders (Deciphering Developmental Disorders Study [DDS]; *Deciphering Developmental Disorders Study, 2017*), and epilepsy (*Heyne et al., 2018*). WGCNA module gene sets were tested for enrichment of these genes using Fisher's test and corrected for multiple comparisons (*Table 1*, *Figure 4a*). Two modules – M12 and M15 – showed enrichment for multiple disease sets, with the ASD gene set being unique for showing enrichment in both modules. We therefore focused downstream analysis on further characterizing the enrichment of ASD genes in M12 and M15, and testing whether these enrichments could predict regional cortical changes in ASD.

## Characterizing ASD gene enrichments in M12 and M15

### kME analysis

To better characterize the spatially distinctive properties of genes within M12 and M15, we defined the union of genes in both modules and collated the WGCNA-defined kME scores for each gene to both M12 and M15. This provided a basis for plotting all genes by their relative membership to both modules to quantify the proximity of each gene to each module, assess the discreteness of gene assignment to modules, and provide a common space within which to project gene functions and associations with ASD (*Figure 4c*).

### Enrichment of ASD-linked GO terms

Genes linked to two specific GO terms, 'Neuronal communication' and 'Gene expression regulation,' enriched among risk genes for ASD in *Satterstrom et al., 2020*, were separately tested for enrichment within M12 and M15 (*Figure 4d*) using Fisher's exact test.

### Developmental trajectories of disease-linked modules

To characterize the distinctive temporal trajectories of M12 and M15 (see *Figure 3c*), we took gene-level trajectories (see 'Developmental trajectories') and calculated the mean gene-expression trajectory of genes in each module (*Figure 4e*).

### Independent characterization of ASD risk genes

To assess the extent to which modules M12 and M15 captured the underlying axes of spatial patterning across all 135 ASD risk genes, we took DEMs for all 135 risk genes and independently clustered them. Pairwise co-expression was calculated for all risk gene DEMs and the resultant matrix was clustered using Gaussian mixture modeling into two clusters, C1 and C2 (*Figure 4—figure supplement 1a*). kME values were calculated for each risk gene with all WGCNA modules and averaged within each cluster. For each cluster, we then identified the WGCNA module with the highest mean kME (*Figure 4—figure supplement 1b*).

## Comparing M12 and M15 expression to regional changes of cortical gene expression in ASD (Figure 4f)

We mapped regional transcriptomic disruption in ASD measured from multiple cortical regions using RNAseq data (*Haney et al., 2020*). This study compared bulk transcriptomic expression in ASD and control samples across 11 cortical areas, quantifying the extent of transcriptomic disruption by identifying the number of significantly DEGs in each region. Cortical areas sampled in this study were mapped to their closest corresponding area in a multimodal MRI parcellation (*Glasser et al., 2016*). The mean expression of M12 and M15 eigenmaps was quantified in the same cortical areas (*Figure 4f*). The test statistic, correlating eigenmap expression with the number of DEGs, was tested against a null distribution generated through spinning and resampling the eigenmaps (see *Table 1*).

## Comparing M12 and M15 expression to regional changes of cortical thickness in ASD (Figure 4g and h, Figure S5c)

To assess the extent to which WGCNA module eigenmaps pattern macroscale in vivo anatomical differences in ASD, we took the map of relative cortical thickness change in autism (see 'Preprocessing and analysis of structural MRI data') and compared this to eigenmap expression patterns. M12 and M15 eigenmaps were thresholded, identifying the 5% of vertices with the highest expression. Areas of high significant thickness change were tested for overlap with areas of significant cortical thickness change using the Dice overlap compared to a null distribution of Dice scores generated through spinning the thresholded eigenmaps (see *Table 1*).

## Preprocessing and analysis of structural MRI data

### AHBA donors

Pial and white matter cortical T1 MRI scans of the six AHBA donor brains were reconstructed using FreeSurfer (v5.3) (*Romero-Garcia et al., 2018*; see *Supplementary file 1*). Briefly, scans undergo tissue segmentation, cortical white and pial surface extraction. A mid-thickness surface between pial and white surfaces was also created. The locations of tissue samples taken for bulk transcriptomic profiling, provided in the coordinates of the subject's MRI, were mapped to the mid-thickness surface as outlined above (see 'Creating spatially dense maps of human cortical gene expression from the AHBA'). Individual subject cortical surfaces were co-registered to the fs_LR32k template surface brain using MSMSulc (*Robinson et al., 2018*) as part of the ciftify pipeline (*Dickie et al., 2019*), which warps subject meshes by nonlinear alignment their folding patterns to the MRI-derived template surface. A donor-specific template surface was created by averaging the coordinates of the aligned meshes and used for analysis of cortical folding patterns used in 'Alignment with reference measures of cortical organization.' Pial, inflated, and flattened representations of the fs_LR32k surface were used for the visualization of cortical maps throughout.

### OASIS (*Figure 1e*)

To estimate relative cortical thickness change in AD patients with the APOE E4 variant, we utilized the openly available OASIS database (*LaMontagne et al., 2019*). T1w MRI data was collected using a Siemens Tim Trio 3T scanner and underwent cortical surface reconstruction using FreeSurfer v5.3 as above. Reconstructions underwent manual quality control and correction, with poor quality data being removed. Output cortical thickness maps, smoothed at 20 mm FWHM and aligned to the fsaverage template surface, were downloaded via https://www.oasis-brains.org/, along with age, sex, and APOE genotype and cognitive status. Subjects were included in the analysis if they had been diagnosed with AD and had at least one APOE E4 allele (n = 119) or were a healthy control (n = 633) (see *Supplementary file 1*). Per-vertex coefficients for disease-associated cortical thinning and significance were calculated, adjusting for age, sex, and mean cortical thickness. We controlled for mean CT to identify local anatomical changes given our finding of generalized cortical thickening in AD compared to controls in OASIS. The map of cortical thickness coefficients was then registered from fsaverage to fs_LR32k for comparison with the DEM of APOE (*Figure 1e*; *Robinson et al., 2018*).

## ABIDE

To estimate relative cortical thickness change in ASD, MRI cortical thickness maps, generated through FreeSurfer processing of 3T T1 structural MRI scans, were downloaded from ABIDE, along with age, sex, and site information (*Di Martino et al., 2017*; *Supplementary file 1*). Multiple sites and scanners were used to acquire these data, which is known to introduce systematic biases in morphological measurements like cortical thickness. To mitigate this, we used neuroCombat, which estimates and removes unwanted scanner effects while retaining biological effects on variables such as age, sex, and diagnosis (*Fortin et al., 2018*). Subjects with poor quality FreeSurfer segmentations were excluded using a threshold Euler count of 100 (ref). Cortical thickness change in ASD relative to controls was calculated adjusting for age, sex, and mean cortical thickness. Neighbor-connected vertices exhibiting significant cortical thickness change (p<0.05) were grouped into clusters. A null distribution of cluster sizes was generated using 1000 random permutations of the cohort, storing the maximum significant cluster size for each permutation. The 95th percentile cluster size was used as a threshold for removing test clusters that could have arisen by chance (*Hagler et al., 2006*). Output coefficient and cluster maps were registered from fsaverage to fs_LR32k and compared with the M12 and M15 eigenmaps as described above.

## Acknowledgements

The authors thank all the participants and their families for their generous involvement in this study. The National Institute of Mental Health Intramural Research Program NIH Annual Report Number, 1ZIAMH002949-04 (AR); Wellcome Trust 215901/Z/19/Z (KSW); NIH grant R01MH112847 (RTS, TDS); NIH grant R01MH123550 (RTS); NIH grant R01MH120482 (RTS, TDS); NIH grant R37MH125829 (TDS); NIH grant R01MH123563 (SNV); NIH grant R01MH120482 (TDS); NIH grant R01EB022573 (TDS); Gates Cambridge Trust (RD); NIH grant T32HG010464 (TTM); MQ: Transforming Mental Health MQF17_24 (PEV); NIH grant T32MH019112 (JS); NIH grant K08MH120564 (AAB, JS); and Rosetrees Trust project grant A2665 (SA).

## Additional information

### Competing interests

Russell T Shinohara: receives consulting income from Octave Bioscience and compensation for reviewership duties from the American Medical Association. The other authors declare that no competing interests exist.

### Funding

| Funder | Grant reference number | Author |
|---|---|---|
| Wellcome Trust | 10.35802/215901 | Konrad Wagstyl |
| National Institute of Mental Health | 1ZIAMH002949-04 | Armin Raznahan |
| National Institutes of Health | T32HG010464 | Jakob Seidlitz |
| MQ Mental Health Research | MQF17_24 | Petra E Vértes |
| Rosetrees Trust | A2665 | Sophie Adler |
| Gates Cambridge Trust | | Richard Dear |
| National Institutes of Health | T32MH019112 | Jakob Seidlitz |
| National Institutes of Health | K08MH120564 | Jakob Seidlitz |
| National Institutes of Health | R01EB022573 | Jakob Seidlitz |

| Funder | Grant reference number | Author |
|---|---|---|
| National Institutes of Health | R01MH120482 | Jakob Seidlitz |
| National Institutes of Health | R01MH123563 | Jakob Seidlitz |
| National Institutes of Health | R37MH125829 | Jakob Seidlitz |
| National Institutes of Health | R01MH120482 | Jakob Seidlitz |
| National Institutes of Health | R01 | Jakob Seidlitz |

The funders had no role in study design, data collection and interpretation, or the decision to submit the work for publication. For the purpose of Open Access, the authors have applied a CC BY public copyright license to any Author Accepted Manuscript version arising from this submission.

## Author contributions

Konrad Wagstyl, Conceptualization, Resources, Data curation, Software, Formal analysis, Funding acquisition, Investigation, Visualization, Methodology, Writing – original draft, Writing – review and editing; Sophie Adler, Formal analysis, Methodology, Writing – original draft, Writing – review and editing; Jakob Seidlitz, Conceptualization, Data curation, Software, Formal analysis, Writing – review and editing; Simon Vandekar, Methodology, Writing – review and editing; Travis T Mallard, Richard Dear, Russell T Shinohara, Aaron Alexander-Bloch, Methodology, Writing – original draft, Writing – review and editing; Alex R DeCasien, Theodore D Satterthwaite, Petra E Vértes, Conceptualization, Methodology, Writing – original draft, Writing – review and editing; Siyuan Liu, Conceptualization, Data curation, Formal analysis, Methodology, Writing – original draft, Writing – review and editing; Daniel H Geschwind, Conceptualization, Supervision, Methodology, Writing – original draft, Writing – review and editing; Armin Raznahan, Conceptualization, Resources, Data curation, Formal analysis, Supervision, Investigation, Visualization, Methodology, Writing – original draft, Writing – review and editing

## Author ORCIDs

Konrad Wagstyl http://orcid.org/0000-0003-3439-5808
Theodore D Satterthwaite http://orcid.org/0000-0001-7072-9399
Siyuan Liu http://orcid.org/0000-0003-3661-6248
Petra E Vértes http://orcid.org/0000-0002-0992-3210
Daniel H Geschwind http://orcid.org/0000-0003-2896-3450
Armin Raznahan http://orcid.org/0000-0002-5622-1190

Reviewer #1 (Public Review): https://doi.org/10.7554/eLife.86933.3.sa1
Reviewer #2 (Public Review): https://doi.org/10.7554/eLife.86933.3.sa2
Author Response https://doi.org/10.7554/eLife.86933.3.sa3

# Additional files

## Supplementary files

• Supplementary file 1. Study participants demographics. (1) Demographics of adult donors Allen Human Brain Atlas microarray data. (2) Demographics of fetal samples from Allen Institute's fetal Laser Microarray Dataset. (3) Demographics of included participants for Alzheimer's disease APOE analysis from the Open Access Series of Imaging Studies (OASIS). (4) Demographics of included participants for autism spectrum disorder analysis from the Autism Brain Imaging Data Exchange (ABIDE) I and II.

• Supplementary file 2. Gene lists used in the study. (1) Gene list assignments for enrichment analyses including WGCNA modules. (2) Meta module assignments.

• Supplementary file 3. Transcriptomically distinctive (TD) peaks. (1) TD genes, GO, cellular, fetal,

and functional annotations. (2) Remaining sheets describe significant Biological Process and Cellular Compartment Gene Ontology annotations for TD peaks.

• Supplementary file 4. WGCNA module enrichments. (1) WGCNA module spatial and gene set enrichment p-values. (2) Remaining sheets describe significant Biological Process and Cellular Compartment Gene Ontology annotations for WGCNA modules.

• MDAR checklist

## Data availability

The cortical dense expression and gradient maps of 20,781 genes and ~30k vertices that support the findings of this study are available at https://rdr.ucl.ac.uk/articles/dataset/MAGICC_vertex-level_gene_expression_data/22183891/1. Scripts to download, visualize, and analyze MAGICC are available at https://github.com/kwagstyl/MAGICC (copy archived at *Wagstyl, 2024*).

The following dataset was generated:

| Author(s) | Year | Dataset title | Dataset URL | Database and Identifier |
|---|---|---|---|---|
| Wagstyl K, Adler S, Vandekar S, Seidlitz J, Mallard TT, Dear R, DeCasien AR, Satterthwaite TD, Liu S, Vertes P, Shinohara RT, Alexander-Bloch A, Geschwind DH, Raznahan A | 2024 | MAGICC vertex-level gene expression data | https://doi.org/10.5522/04/22183891.v1 | University College London, 10.5522/04/22183891.v1 |

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
