## [Editor Report · eLife assessment]

This study provides continuous maps of human brain gene expression and explores their relationship with a large variety of microscopic and macroscopic aspects of brain organisation. The authors provide **convincing** evidence for a relationship between gene expression maps with various aspects of the anatomy of adult brains, during development, and in the case of mental disorders. The data and methods introduced can be an **important** tool for neuroimaging research.

---

## [Referee Report · Reviewer #1 (Public Review)]

The manuscript by Wagstyl et al. describes an extensive analysis of gene expression in the human cerebral cortex and the association with a large variety of maps capturing many of its microscopic and macroscopic properties. The core methodological contribution is the computation of continuous maps of gene expression for >20k genes, which are being shared with the community. The manuscript is a demonstration of several ways in which these maps can be used to relate gene expression with histological features of the human cortex, cytoarchitecture, folding, function, development and disease risk. The main scientific contribution is to provide data and tools to help substantiate the idea of the genetic regulation of multi-scale aspects of the organisation of the human brain. The manuscript is dense, but clearly written and beautifully illustrated.

---

## [Referee Report · Reviewer #2 (Public Review)]

This is a valuable contribution that will facilitate brain transcriptomic analyses and the joint analyses of gene expression and structural and functional imaging. The methods used are solid, and the authors conducted a wide range of analyses to demonstrate the value of the dense gene expression data.

---

## [Author Response]

The following is the authors’ response to the original reviews.

**Reviewer #1 (Public Review):**
The manuscript by Wagstyl et al. describes an extensive analysis of gene expression in the human cerebral cortex and the association with a large variety of maps capturing many of its microscopic and macroscopic properties. The core methodological contribution is the computation of continuous maps of gene expression for >20k genes, which are being shared with the community. The manuscript is a demonstration of several ways in which these maps can be used to relate gene expression with histological features of the human cortex, cytoarchitecture, folding, function, development and disease risk. The main scientific contribution is to provide data and tools to help substantiate the idea of the genetic regulation of multi-scale aspects of the organisation of the human brain. The manuscript is dense, but clearly written and beautifully illustrated.Main commentsThe starting point for the manuscript is the construction of continuous maps of gene expression for most human genes. These maps are based on the microarray data from 6 left human brain hemispheres made available by the Allen Brain Institute. By technological necessity, the microarray data is very sparse: only 1304 samples to map all the cortex after all subjects were combined (a single individual's hemisphere has ~400 samples). Sampling is also inhomogeneous due to the coronal slicing of the tissue. To obtain continuous maps on a mesh, the authors filled the gaps using nearest-neighbour interpolation followed by strong smoothing. This may have two potentially important consequences that the authors may want to discuss further: (a) the intrinsic geometry of the mesh used for smoothing will introduce structure in the expression map, and (b) strong smoothing will produce substantial, spatially heterogeneous, autocorrelations in the signal, which are known to lead to a significant increase in the false positive rate (FPR) in the spin tests they used.

Many thanks to the reviewer for their considered feedback. We have addressed these primary concerns into point-by-point responses below. The key conclusions from our new analyses are: (i) while the intrinsic geometry of the mesh had not originally been accounted for in sufficient detail, the findings presented in this manuscript paper are not driven by mesh-induced structure, (ii) that the spin test null models used in this manuscript [including a modified version introduced in response to (i)] are currently the most appropriate way to mitigate against inflated false positive rates when making statistical inferences on smooth, surface-based data.

a. Structured smoothingA brain surface has intrinsic curvature (Gaussian curvature, which cannot be flattened away without tearing). The size of the neighbourhood around each surface vertex will be determined by this curvature. During surface smoothing, this will make that the weight of each vertex will be also modulated by the local curvature, i.e., by large geometric structures such as poles, fissures and folds. The article by Ciantar et al (2022, https://doi.org/10.1007/s00429-022-02536-4) provides a clear illustration of this effect: even the mapping of a volume of *pure noise* into a brain mesh will produce a pattern over the surface strikingly similar to that obtained by mapping resting state functional data or functional data related to a motor task.Comment 1It may be important to make the readers aware of this possible limitation, which is in large part a consequence of the sparsity of the microarray sampling and the necessity to map that to a mesh. This may confound the assessments of reproducibility (results, p4). Reproducibility was assessed by comparing pairs of subgroups split from the total 6. But if the mesh is introducing structure into the data, and if the same mesh was used for both groups, then what's being reproduced could be a combination of signal from the expression data and signal induced by the mesh structure.

Response 1

The reviewer raises an important question regarding the potential for interpolation and smoothing on a cortical mesh to induce a common/correlated signal due to the intrinsic mesh structure. We have now generated a new null model to test this idea which indicates that intrinsic mesh structure is not inflating reproducibility in interpolated expression maps. This new null model spins the original samples prior to interpolation, smoothing and comparison between triplet splits of the six donors, with independent spins shared across the triplet. For computational tractability we took one pair of triplets and regenerated the dataset for each triplet using 10 independent spins. We used these to estimate gene-gene null reproducibility for 90 independent pairwise combinations of these 10 spins. Across these 90 permutations, the average median gene-gene correlation was R=0.03, whereas in the unspun triplet comparisons this was R=0.36. These results indicate that the primary source of the gene-level triplet reproducibility is the underlying shared gene expression pattern rather than interpolation-induced structure.

In Methods 2a: "An additional null dataset was generated to test whether intrinsic geometry of the cortical mesh and its impact on interpolation for benchmarking analyses of DEMs and gradients (Fig S1d, Fig S2d, Fig S3c). In these analyses, the original samples were rotated on the spherical surface prior to subsequent interpolation, smoothing and gradient calculation. Due to computational constraints the full dataset was recreated only for 10 independent spins. These are referred to as the “spun+interpolated null”.

**Author response image 1. sa3fig1:** Figure S1d: Gene predictability was higher across all triplet-triplet pairs than when compared to spun+interpolated null.

Comment 2It's also possible that mesh-induced structure is responsible in part for the "signal boost" observed when comparing raw expression data and interpolated data (fig S1a). How do you explain the signal boost of the smooth data compared with the raw data otherwise?

Response 2

We thank the reviewer for highlighting this issue of mesh-induced structure. We first sought to quantify the impact of mesh-induced structure through the new null model, in which the data are spun prior to interpolation. New figure S1d, S2d and S3c all show that the main findings are not driven by interpolation over a common mesh structure, but rather originate in the underlying expression data.

Specifically, for the original Figure S1a, the reviewer highlights a limitation that we compared intersubject predictability of raw-sample to raw-sample and interpolated-to-interpolated. In this original formulation improved prediction scores for interpolated-to-interpolated (the “signal boost”) could be driven by mesh-induced structure being applied to both the input and predicted maps. We have updated this so that we are now comparing raw-to-raw and interpolated-to-raw, i.e. whether interpolated values are better estimations of the measured expression values. The new Fig S1a&b (see below) shows a signal boost in gene-level and vertex level prediction scores (delta R = +0.05) and we attribute this to the minimisation of location and measurement noise in the raw data, improving the intersubject predictability of expression levels.

In Methods 2b: "To assess the effect of data interpolation in DEM generation we compared gene-level and vertex-level reproducibility of DEMs against a “ground truth” estimate of these reproducibility metrics based on uninterpolated expression data. To achieve a strict comparison of gene expression values between different individuals at identical spatial locations we focused these analyses on the subset of AHBA samples where a sample from one subject was within 3 mm geodesic distance of another. This resulted in 1097 instances (spatial locations) with measures of raw gene expression of one donor, and predicted values from the second donor’s un-interpolated AHBA expression data and interpolated DEM. We computed gene-level and vertex-level reproducibility of expression using the paired donor data at each of these sample points for both DEM and uninterpolated AHBA expression values. By comparing DEM reproducibility estimates with those for uninterpolated AHBA expression data, we were able to quantify the combined effect of interpolation and smoothing steps in DEM generation. We used gene-level reproducibility values from DEMs and uninterpolated AHBA expression data to compute a gene-level difference in reproducibility, and we then visualized the distribution of these difference values across genes (Fig S1a). We used gene-rank correlation to compare vertex-level reproducibility values between DEMs and uninterpolated AHBA expression data (Fig S1b)."

**Author response image 2. sa3fig2:** Figure S1: Reproducibility of Dense Expression Maps (DEMs) interpolated from spatially sparse postmortem measures of cortical gene expression. (**a**) Signal boost in the interpolated DEM dataset vs. spatially sparse expression data. Restricting to samples taken from approximately the same cortical location in pairs of individuals (within 3mm geodesic distance), there was an overall improvement in intersubject spatial predictability in the interpolated maps. Furthermore, genes with lower predictability in the interpolated maps were less predictable in the raw dataset, suggesting these regions exhibit higher underlying biological variability rather than methodologically introduced bias. (**b**) Similarly at the paired sample locations, gene-rank predictability was generally improved in DEMs vs. sparse expression data (median change in R from sparse samples to interpolated for each pair of subjects, +0.5).

1. How do you explain that despite the difference in absolute value the combined expression maps of genes with and without cortical expression look similar? (fig S1e: in both cases there's high values in the dorsal part of the central sulcus, in the occipital pole, in the temporal pole, and low values in the precuneus and close to the angular gyrus). Could this also reflect mesh-smoothing-induced structure?

Response 3

As with comment 1, this is an interesting perspective that we had not fully considered. We would first like to clarify that non-cortical expression is defined from the independent datasets including the “cortex” tissue class of the human protein atlas and genes identified as markers for cortical layers or cortical cells in previous studies. This is still likely an underestimate of true cortically expressed genes as some of these “non-cortical genes” had high intersubject reproducibility scores. Nevertheless we think it appropriate to use a measure of brain expression independent of anything included in other analyses for this paper. These considerations are part of the reason we provide all gene maps with accompanying uncertainty scores for user discretion rather than simply filtering them out.

In terms of the spatially consistent pattern of the gene ranks of Fig S1f, this consistent spatial pattern mirrors Transcriptomic Distinctiveness (r=0.52 for non-cortical genes, r=0.75 for cortical genes), so we think that as the differences in expression signatures become more extreme, the relative ranks of genes in that region are more reproducible/easier to predict.

To assess whether mesh-smoothing-induced structure is playing a role, we carried out an additional the new null model introduced in response to comment 1, and asked if the per-vertex gene rank reproducibility of independently spun subgroup triplets showed a similar structure to that in our original analyses. Across the 90 permutations, the median correlation between vertex reproducibility and TD was R=0.10. We also recalculated the TD maps for the 10 spun datasets and the mean correlation with the original TD did not significantly differ from zero (mean R = 0.01, p=0.2, nspins = 10). These results indicate that folding morphology is not the major driver of local or large scale patterning in the dataset. We have included this as a new Figure S3c.

We have updated the text as follows:

In Methods 3a: "Third, to assess whether the covariance in spatial patterning across genes could be a result of mesh-associated structure introduced through interpolation and smoothing, TD maps were recomputed for the spun+interpolated null datasets and compared to the original TD map (Fig S3c)."

In Results: "The TD map observed from the full DEMs library was highly stable between all disjoint triplets of donors (Methods, Fig S3a, median cross-vertex correlation in TD scores between triplets r=0.77) and across library subsets at all deciles of DEM reproducibility (Methods, Fig S3b, cross-vertex correlation in TD scores r>0.8 for the 3rd-10th deciles), but was not recapitulated in spun null datasets (Fig S3c)."

**Author response image 3. sa3fig3:** Figure S3c: Correlations between TD and TD maps regenerated on datasets spun using two independent nulls, one where the rotation is applied prior to interpolation and smoothing (spun+interpolated) and one where it is applied to the already-created DEMs. In each null, the same rotation matrix is applied to all genes.

Comment 4Could you provide more information about the way in which the nearest-neighbours were identified (results p4). Were they nearest in Euclidean space? Geodesic? If geodesic, geodesic over the native brain surface? over the spherically deformed brain? (Methods cite Moresi & Mather's Stripy toolbox, which seems to be meant to be used on spheres). If the distance was geodesic over the sphere, could the distortions introduced by mapping (due to brain anatomy) influence the geometry of the expression maps?

Response 4

We have clarified in the Methods that the mapping is to nearest neighbors on the spherically-inflated surface.

The new null model we have introduced in response to comments 1 & 3 preserves any mesh-induced structure alongside any smoothing-induced spatial autocorrelations, and the additional analyses above indicate that main results are not induced by systematic mesh-related interpolation signal. In response to an additional suggestion from the reviewer (Comment 13), we also assessed whether local distortions due to the mesh could be creating apparent border effects in the data, for instance at the V1-V2 boundary. At the V1-V2 border, which coincides anatomically with the calcarine sulcus, we computed the 10 genes with the highest expression gradient along this boundary in the actual dataset and the spun-interpolated null. The median test expression gradients along this border was higher than in any of the spun datasets, indicating that these boundary effects are not explained by the interpolation and cortical geometry effects on the data (new Fig S2d).The text has been updated as follows:

In Methods 1: "For cortical vertices with no directly sampled expression, expression values were interpolated from their nearest sampled neighbor vertex on the spherical surface (Moresi and Mather, 2019) (Fig 1b)."

In Methods 2: "We used the spun+interpolated null to test whether high gene gradients could be driven by non-uniform interpolation across cortical folds. We quantified the average gradient for all genes along the V1-V2 border in the atlas, as well as for 10 iterations of the atlas where the samples were spun prior to interpolation. We computed the median gradient magnitude for the 20 top-ranked genes for each (Fig S2d)."

**Author response image 4. sa3fig4:** Figure S2d: Mean of gradient magnitudes for 20 genes with largest gradients along V1-V2 border, compared to values along the same boundary on the spun+interpolated null atlas. Gradients were higher in the actual dataset than in all spun version indicating this high gradient feature is not primarily due to the effects of calcarine sulcus morphology on interpolation

Comment 5Could you provide more information about the smoothing algorithm? Volumetric, geodesic over the native mesh, geodesic over the sphere, averaging of values in neighbouring vertices, cotangent-weighted laplacian smoothing, something else?

Response 5

We are using surface-based geodesic over the white surface smoothing described in Glasser et al., 2013 and used in the HCP workbench toolbox (https://www.humanconnectome.org/software/connectome-workbench). We have updated the methods to clarify this.

In Methods 1: "Surface expression maps were smoothed using the Connectome Workbench toolbox (Glasser et al. 2013) with a 20mm full-width at half maximum Gaussian kernel , selected to be consistent with this sampling density (Fig 1c)."

Comment 6Could you provide more information about the method used for computing the gradient of the expression maps (p6)? The gradient and the laplacian operator are related (the laplacian is the divergence of the gradient), which could also be responsible in part for the relationships observed between expression transitions and brain geometry.

Response 6

We are using Connectome Workbench’s metric gradient command for this Glasser et al., 2013 and used in the HCP workbench pipeline. The source code for gradient calculation can be found here:https://github.com/Washington-University/workbench/blob/131e84f7b885d82af76e be21adf2fa97795e2484/src/Algorithms/AlgorithmMetricGradient.cxx

In Methods 2: >For each of the resulting 20,781 gene-level expression maps, the orientation and magnitude of gene expression change at each vertex (i.e. the gradient) was calculated for folded, inflated, spherical and flattened mesh representations of the cortical sheet using Connectome Workbench’s metric gradient command (Glasser et al. 2013).

b. Potentially inflated FPR for spin tests on autocorrelated data."Spin tests are extensively used in this work and it would be useful to make the readers aware of their limitations, which may confound some of the results presented. Spin tests aim at establishing if two brain maps are similar by comparing a measure of their similarity over a spherical deformation of the brains against a distribution of similarities obtained by randomly spinning one of the spheres. It is not clear which specific variety of spin test was used, but the original spin test has well known limitations, such as the violation of the assumption of spatial stationarity of the covariance structure (not all positions of the spinning sphere are equivalent, some are contracted, some are expanded), or the treatment of the medial wall (a big hole with no data is introduced when hemispheres are isolated).Another important limitation results from the comparison of maps showing autocorrelation. This problem has been extensively described by Markello & Misic (2021). The strong smoothing used to make a continuous map out of just ~1300 samples introduces large, geometry dependent autocorrelations. Indeed, the expression maps presented in the manuscript look similar to those with the highest degree of autocorrelation studied by Markello & Misic (alpha=3). In this case, naive permutations should lead to a false positive rate ~46% when comparing pairs of random maps, and even most sophisticated methods have FPR>10%.Comment 7There's currently several researchers working on testing spatial similarity, and the readers would benefit from being made aware of the problem of the spin test and potential solutions. There's also packages providing alternative implementations of spin tests, such as BrainSMASH and BrainSpace, which could be mentioned.

Response 7

We thank the reviewer for raising the issue of null models. First, with reference to the false positive rate of 46% when maps exhibit spatial autocorrelation, we absolutely agree that this is an issue that must be accounted for and we address this using the spin test. We acknowledge there has been other work on nulls such as BrainSMASH and BrainSpace. Nevertheless in the Markello and Misic paper to which the reviewer refers, the BrainSmash null models perform worse with smoother maps (with false positive rates approaching 30% in panel e below), whereas the spin test maintains false positives rates below 10%.

**Author response image 5. sa3fig5:** 

We have added a brief description of the challenge and our use of the spin test.

In Methods 2a: "Cortical maps exhibit spatial autocorrelation that can inflate the False Positive Rate, for which a number of methods have been proposed(Alexander-Bloch et al. 2018; Burt et al. 2020; Vos de Wael et al. 2020). At higher degrees of spatial smoothness, this high False Positive Rate is most effectively mitigated using the spin test(Alexander-Bloch et al. 2018; Markello and Misic 2021; Vos de Wael et al. 2020). In the following analyses when generating a test statistic comparing two spatial maps, to generate a null distribution, we computed 1000 independent spins of the cortical surface using https://netneurotools.readthedocs.io, and applied it to the first map whilst keeping the second map unchanged. The test statistic was then recomputed 1000 times to generate a null distribution for values one might observe by chance if the maps shared no common organizational features. This is referred to throughout as the “spin test” and the derived p-values as pspin."

Comment 8Could it be possible to measure the degree of spatial autocorrelation?

Response 8

We agree this could be a useful metric to generate for spatial cortical maps. However, there are multiple potential metrics to choose from and each of the DEMs would have their own value. To address this properly would require the creation of a set of validated tools and it is not clear how we could summarize this variety of potential metrics for 20k genes. Moreover, as discussed above the spin method is an adequate null across a range of spatial autocorrelation degrees, thus while we agree that in general estimation of spatial smoothness could be a useful imaging metric to report, we consider that it is beyond the scope of the current manuscript.

Comment 9Could you clarify which version of the spin test was used? Does the implementation come from a package or was it coded from scratch?

Response 9

As Markello & Misic note, at the vertex level, the various implementations of the spin test become roughly equivalent to the ‘original’ Alexander-Bloch et al., implementation. We used took the code for the ‘original’ version implemented in python here:https://netneurotools.readthedocs.io/en/latest/_modules/netneurotools/stats.html# gen_spinsamples.

This has been updated in the methods (see Response 7).

Comment 10Cortex and non-cortex vertex-level gene rank predictability maps (fig S1e) are strikingly similar. Would the spin test come up statistically significant? What would be the meaning of that, if the cortical map of genes not expressed in the cortex appeared to be statistically significantly similar to that of genes expressed in the cortex?

Response 10

Please see response to comment 3, which also addresses this observation.

**Reviewer #2 (Public Review):**
The authors convert the AHBA dataset into a dense cortical map and conduct an impressively large number of analyses demonstrating the value of having such data.I only have comments on the methodology.Comment 1First, the authors create dense maps by simply using nearest neighbour interpolation followed by smoothing. Since one of the main points of the paper is the use of a dense map, I find it quite light in assessing the validity of this dense map. The reproducibility values they calculate by taking subsets of subjects are hugely under-powered, given that there are only 6 brains, and they don't inform on local, vertex-wise uncertainties. I wonder if the authors would consider using Gaussian process interpolation. It is really tailored to this kind of problem and can give local estimates of uncertainty in the interpolated values. For hyperparameter tuning, they could use leave-one-brain-out for that.I know it is a lot to ask to change the base method, as that means re-doing all the analyses. But I think it would strengthen the paper if the authors put as much effort in the dense mapping as they did in their downstream analyses of the data.

Response 1

We thank the reviewer for the suggestion to explore Gaussian process interpolation. We have implemented this for our dataset and attempted to compare this with our original method with the 3 following tests: (i) intertriplet reproducibility of individual gene maps, (ii) microscale validations: area markers, (iii) macroscale validations: bio patterns.

Overall, compared to our original nearest-neighbor interpolation method, GP regression (i) did not substantially improve gene-level reproducibility of expression maps (median correlation increase of R=0.07 which was greater for genes without documented protein expression in cortex): (ii) substantially worsened performance in predicting areal marker genes and (iii) showed similar but slightly worse performance at predicting macroscale patterns from Figure 1.

Given the significantly poorer performance on one of our key tests (ii) we have opted not to replace our original database, but we do now include code for the alternative GP regression methodology in the github repository so others can reproduce/further develop these methods.

**Author response image 6. sa3fig6:** 

ii) Genes ranked by mean expression gradient from current DEMs (left) and Gaussian process-derived interpolation maps (right). Established Human and macaque markers are consistently higher-ranked in DEM maps. iii) Figure 1 Interpolated vs GP regression

**Author response table 1. sa3table1:** 

	Interpolated	GP regression
Fig 1g Cell types	R=0.62	R=0.59
Fig 1i L4	R=0.77	R=0.63
Fig 1i Betz	Delta z=1.76	Delta z=1.31
Fig 1i Myelin	R=0.60	R=0.61
Fig 1i APOE	R=-0.52	R=-0.48

Comment 2It is nice that the authors share some code and a notebook, but I think it is rather light. It would be good if the code was better documented, and if the user could have access to the non-smoothed data, in case they was to produce their own dense maps. I was only wondering why the authors didn't share the code that reproduces the many analyses/results in the paper.

Response 2

We thank the reviewer for this suggestion. In response we have updated the shared github repository (https://github.com/kwagstyl/magicc). This now includes code and notebooks to reproduce the main analyses and figures.

**Reviewer #1 (Recommendations For The Authors):**
Minor commentsComment 11p4 mentions Fig S1h, but the supp figures only goes from S1a to S1g

Response 11

We thank the reviewer for capturing this error. It was in fact referring to what is now Fig S1h and has been updated.

Comment 12It would be important that the authors share all the code used to produce the results in the paper in addition to the maps. The core methodological contribution of the work is a series of continuous maps of gene expression, which could become an important tool for annotation in neuroimaging research. Many arbitrary (reasonable) decisions were made, it would be important to enable users to evaluate their influence on the results.

Response 12

We thank both reviewers for this suggestion. We have updated the github to be able to reproduce the dense maps and key figures with our methods.

Comment 13p5: Could the sharp border reflect the effect of the geometry of the calcarine sulcus on map smoothing? More generally, could there be an effect of folds on TD?

Response 13

Please see our response to Reviewer 1, Comment 1 above, where we introduce the new null models now analyzed to test for effects of mesh geometry on our findings. These new null models - where original source data were spun prior to interpolation suggest that neither the sharp V1/2 border or the TD map are effects of mesh geometry. Specifically: (i) , the magnitudes of gradients along the V1/2 boundary from null models were notably smaller than those in our original analyses (see new figure S2d), and (ii) TD maps computed from the new null models showed no correlation with TD maps from ur original analyses (new Figure S3c, mean R = 0.01, p=0.2, nspins = 10).

Comment 14p5: Similar for the matching with the areas in Glasser's parcellation: the definition of these areas involves alignment through folds (based on freesurfer 'sulc' map, see Glasser et al 2016). If folds influence the geometry of TDs, could that influence the match?

Response 14

We note that Fig S3c provided evidence that folding was not the primary driver of the TD patterning. However, it is true that Glasser et al. use both neuroanatomy (folding, thickness and myelin) and fMRI-derived maps to delineate their cortical areas. As such Figure 2 f & g aren’t fully independent assessments. Nevertheless the reason that these features are used is that many of the sulci in question have been shown to reliably delineate cytoarchitectonic boundaries (Fischl et al., 2008).

In Results: "A similar alignment was seen when comparing gradients of transcriptional change with the spatial orientation of putative cortical areas defined by multimodal functional and structural in vivo neuroimaging(Glasser et al., 2016) (expression change running perpendicular to area long-axis, pspin<0.01, Fig 2g, Methods)."

Comment 15p6: TD peaks are said to overlap with functionally-specialised regions. A comment on why audition is not there, nor language, but ba 9-46d is? Would that suggest a lesser genetic regulation of those functions?

Response 15

The reviewer raises a valid point and this was a result that we were also surprised by. The finding that the auditory cortex is not as microstructurally distinctive as, say V1, is consistent with other studies applying dimensionality-reduction techniques to multimodal microstructural receptor data (e.g. Zilles et al., 2017, Goulas et al., 2020). These studies found that the auditory microstructure is not as extreme as either visual and somatomotor areas. From a methodological view point, the primary auditory cortex is significantly smaller than both visual and somatomotor areas, and therefore is captured by fewer independent samples, which could reduce the detail in which its structure is being mapped in our dataset.

For the frontal areas, we would note that (i) the frontal peak is the smallest of all peaks found and was more strongly characterised by low z-score genes than high z-score. (ii) the anatomical areas in the frontal cortex are much more highly variable with respect to folding morphology (e.g. Rajkowska 1995). The anatomical label of ba9-46d (and indeed all other labels) were automatically generated as localisers rather than strict area labels. We have clarified this in the text as follows:

In Methods 3a: "Automated labels to localize TD peaks were generated based on their intersection with a reference multimodal neuroimaging parcellation of the human cortex(Glasser et al., 2016). Each TD was given the label of the multimodal parcel that showed greatest overlap (Fig 2b)."

Comment 16.p7: The proposition that "there is a tendency for cortical sulci to run perpendicular to the direction of fastest transcriptional change", could also be "there is a tendency for the direction of fastest transcriptional change to run perpendicular to cortical sulci"? More pragmatically, this result from the geometry of transcriptional maps being influenced by sulcal geometry in their construction.

Response 16

Please see our response to Reviewer 1, Comment 1 above, where we introduce the new null models now analyzed to test for effects of mesh geometry on our findings. These models indicate that the topography of interpolated gene expression maps do not reflect influences of sulcal geometry on their construction.

Comment 17p7: TD transitions are indicated to precede folding. This is based on a consideration of folding development based on the article by Chi et al 1977, which is quite an old reference. In that paper, the authors estimated the tempo of human folding development based on the inspection of photographs, which may not be sufficient for detecting the first changes in curvature leading to folds. The work of the Developing Human Connectome consortium may provide a more recent indication for timing. In their data, by PCW 21 there's already central sulcus, pre-central, post-central, intra-parietal, superior temporal, superior frontal which can be detected by computing the mean curvature of the pial surface (I can only provide a tweet for reference: https://twitter.com/R3RT0/status/1617119196617261056). Even by PCW 9-13 the callosal sulcus, sylvian fissure, parieto-occipital fissure, olfactory sulcus, cingulate sulcus and calcarine fissure have been reported to be present (Kostovic & Vasung 2009).

Response 17

Our field lacks the data necessary to provide a comprehensive empirical test for the temporal ordering of regional transcriptional profiles and emergence of folding. Our results show that transcriptional identities of V1 and TGd are - at least - present at the very earliest stages of sulcation in these regions. In response to the reviewers comment we have updated with a similar fetal mapping project which similarly shows evidence of the folds between weeks 17-21 and made the language around directionality more cautious.

In Results: "The observed distribution of these angles across vertices was significantly skewed relative to a null based on random alignment between angles (pspin<0.01, Fig 2f, Methods) - indicating that there is indeed a tendency for cortical sulci and the direction of fastest transcriptional change to run perpendicular to each other (pspin<0.01, Fig 2f).

As a preliminary probe for causality, we examined the developmental ordering of regional folding and regional transcriptional identity. Mapping the expression of high-ranking TD genes in fetal cortical laser dissection microarray data(Miller et al., 2014) from 21 PCW (Post Conception Weeks) (Methods) showed that the localized transcriptional identity of V1 and TGd regions in adulthood is apparent during the fetal periods when folding topology begins to emerge (Chi et al. 1977; Xu et al. 2022) (Fig "S2d).

In Discussion: "By establishing that some of these cortical zones are evident at the time of cortical folding, we lend support to a “protomap”(Rakic 1988; O'Leary 1989; O'Leary et al. 2007; Rakic et al. 2009) like model where the placement of some cortical folds is set-up by rapid tangential changes in cyto-laminar composition of the developing cortex(Ronan et al., 2014; Toro and Burnod, 2005; Van Essen, 2020). The DEMs are derived from fully folded adult donors, and therefore some of the measured genetic-folding alignment might also be induced by mechanical distortion of the tissue during folding(Llinares-Benadero and Borrell 2019; Heuer and Toro 2019). However, no data currently exist to conclusively assess the directionality of this gene-folding relationship."

Comment 18p7: In my supplemental figures (obtained from biorxiv, because I didn't find them among the files submitted to eLife) there's no S2j (only S2a-S2i).

Response 18

We apologize, this figure refers to S3k (formerly S3j), rather than S2j. We have updated the main text.

Comment 19 p7: It is not clear from the methods (section 3b) how the adult and fetal brains were compared. Maybe using MSM (Robinson et al 2014)?

Response 19

We have now clarified this in Methods text as reproduced below.

In Methods 3b: "We averaged scaled regional gene expression values between donors per gene, and filtered for genes in the fetal LDM dataset that were also represented in the adult DEM dataset - yielding a single final 20,476*235 gene-by-sample matrix of expression values for the human cortex at 21 PCW. Each TD peak region was then paired with the closest matching cortical label within the fetal regions. This matrix was then used to test if each TD expression signature discovered in the adult DEM dataset (Fig 2, Table 3) was already present in similar cortical regions at 21 PCW."

Comment 20p7: WGCNA is used prominently, could you provide a brief introduction to its objectives? The gene coexpression networks are produced after adjusting the weight of the network edges to follow a scale-free topology, which is meant to reflect the nature of protein-protein interactions. Soft thresholding increases contrast, but doesn't this decrease a potential role of infinitesimal regulatory signals?

Response 20

We agree with the reviewer that the introduction to WGCNA needed additional details and have amended the Results (see below). One limitation of WGCNA-derived associations is that it will downweigh the role of smaller relationships including potentially important regulatory signals. WGCNA methods have been titrated to capture strong relationships. This is an inherent limitation of all co-expression driven methods which lead to an incomplete characterisation of the molecular biology. Nevertheless we feel these stronger relationships are still worth capturing and interrogating. We have updated the text to introduce WGCNA and acknowledge this potential weakness in the approach.

In Results: "Briefly, WGCNA constructs a constructs a connectivity matrix by quantifying pairwise co-expression between genes, raising the correlations to a power (here 6) to emphasize strong correlations while penalizing weaker ones, and creating a Topological Overlap Matrix (TOM) to capture both pairwise similarities expression and connectivity. Modules of highly interconnected genes are identified through hierarchical clustering. The resultant WGCNA modules enable topographic and genetic integration because they each exist as both (i) a single expression map (eigenmap) for spatial comparison with neuroimaging data (Fig 3a,b, Methods) and, (ii) a unique gene set for enrichment analysis against marker genes systematically capturing multiple scales of cortical organization, namely: cortical layers, cell types, cell compartments, protein-protein interactions (PPI) and GO terms (Methods, Table S2 and S4)."

Comment 21WGCNA modules look even more smooth than the gene expression maps. Are these maps comparable to low frequency eigenvectors? Autocorrelation in that case should be very strong?

Response 21

These modules are smooth as they are indeed eigenvectors which likely smooth out some of the more detailed but less common features seen in individual gene maps. These do exhibit high degrees of autocorrelation, nevertheless we are applying the spin test which is currently the appropriate null model for spatially autocorrelated cortical maps (Response 7).

Comment 22If the WGCNA modules provide an orthogonal basis for surface data, is it completely unexpected that some of them will correlate with low-frequency patterns? What would happen if random low frequency patterns were generated? Would they also show correlations with some of the 16 WGCNA modules?

Response 22

We agree with the reviewer that if we used a generative model like BrainSMASH, we would likely see similar low frequency patterns. However, the inserted figure in Response 7 from Makello & Misic provide evidence that is not as conservative a null as the spin test when data exhibit high spatial autocorrelation. The spatial enrichment tests carried out on the WGCNA modules are all carried out using the spin test.

Comment 23In part (a) I commented on the possibility that brain anatomy may introduce artifactual structure into the data that's being mapped. But what if the relationship between brain geometry and brain organisation were deeper than just the introduction of artefacts? The work of Lefebre et al (2014, https://doi.org/10.1109/ICPR.2014.107; 2018, https://doi.org/10.3389/fnins.2018.00354) shows that clustering based on the 3 lowest frequency eigenvectors of the Laplacian of a brain hemisphere mesh produce an almost perfect parcellation into lobes, with remarkable coincidences between parcel boundaries and primary folds and fissures. The work of Pang et al(https://doi.org/10.1101/2022.10.04.510897) suggests that the geometry of the brain plays a critical role in constraining its dynamics: they analyse >10k task-evoked brain maps and show that the eigenvectors of the brain laplacian parsimoniously explain the activity patterns. Could brain anatomy have a downward effect on brain organisation?

Response 23

The reviewer raises a fascinating extension of our work identifying spatial modes of gene expression. We agree that these are low frequency in nature, but would first like to note that the newly introduced null model indicates that the overlaps with salient neuroanatomical features are inherent in the expression data and not purely driven by anatomy in a methodological sense.

Nevertheless we absolutely agree there is likely to be a complex multidirectional interplay between genetic expression patterns through development, developing morphology and the “final” adult topography of expression, neuroanatomical and functional patterns.

We think that the current manuscript currently contains a lot of in depth analyses of these expression data, but agree that a more extensive modeling analysis of how expression might pattern or explain functional activation would be a fascinating follow on, especially in light of these studies from Pang and Lefebre. Nevertheless we think that this must be left for a future modeling paper integrating these modes of microscale, macroscale and functional anatomy.

In Discussion: "Indeed, future work might find direct links between these module eigenvectors and similar low-frequency eigenvectors of cortical geometry have been used as basis functions to segment the cortex (Lefèvre et al. 2018) and explain complex functional activation patterns(Pang et al. 2023)."

Comment 24On p11: ASD related to rare, deleterious mutations of strong effect is often associated with intellectual disability (where the social interaction component of ASD is more challenging to assess). Was there some indication of a relationship with that type of cognitive phenotype?

Response 24

Across the two ABIDE cohorts, the total number of those with ASD and IQ <70, which is the clinical threshold for intellectual disability was n=10, which unfortunately did not allow us to conduct a meaningful test of whether ID impacts the relationship between imaging changes in ASD and the expression maps of genes implicated in ASD by rare variants.

Comment 25Could you clarify if the 6 donors were aligned using the folding-based method in freesurfer?

Response 25

The 6 donors were aligned using MSMsulc (Robinson et al., 2014), which is a folding based method from the HCP group. This is now clarified in the methods.

In Methods 1: "Cortical surfaces were reconstructed for each AHBA donor MRI using FreeSurfer(Fischl, 2012), and coregistered between donors using surface matching of individuals’ folding morphology (MSMSulc) (Robinson et al., 2018)."

Comment 26The authors make available a rich resource and a series of tools to facilitate their use. They have paid attention to encode their data in standard formats, and their code was made in Python using freely accessible packages instead of proprietary alternatives such as matlab. All this should greatly facilitate the adoption of the approach. I think it would be important to state more explicitly the conceptual assumptions that the methodology brings. In the same way that a GWAS approach relies on a Mendelian idea that individual alleles encode for phenotypes, what is the idea about the organisation of the brain implied by the orthogonal gene expression modules? Is it that phenotypes - micro and macro - are encoded by linear combinations of a reduced number of gene expression patterns? What would be the role of the environment? The role of non-genic regulatory regions? Some modalities of functional organisation do not seem to be encoded by the expression of any module. Is it just for lack of data or should this be seen as the sign for a different organisational principle? Likewise, what about the aspects of disorders that are not captured by expression modules? Would that hint, for example, to stronger environmental effects? What about linear combinations of modules? Nonlinear? Overall, the authors adopt implicitly, en passant, a gene-centric conceptual standpoint, which would benefit from being more clearly identified and articulated. There are citations to Rakic's protomap idea (I would also cite the original 1988 paper, and O'Leary's 1989 "protocortex" paper stressing the role of plasticity), which proposes that a basic version of brain cytoarchitecture is genetically determined and transposed from the proliferative ventricular zone regions to the cortical plate through radial migration. In p13 the authors indicate that their results support Rakic's protomap. Additionally, in p7 the authors suggest that their results support a causal arrow going from gene expression to sulcal anatomy. The reviews by O'leary et al (2007), Ronan & Fletcher (2014, already cited), Llinares-Benadero & Borrell (2019) could be considered, which also advocate for a similar perspective. For nuances on the idea that molecular signals provide positional information for brain development, the article by Sharpe (2019, DOI: 10.1242/dev.185967) is interesting. For nuances on the gene-centric approach of the paper the articles by Rockmann (2012, DOI: 10.1111/j.1558-5646.2011.01486.x) but also from the ENCODE consortium showing the importance of non-genic regions of the genome ("Perspectives on ENCODE" 2020 DOI: 10.1038/s41586-021-04213-8) could be considered. I wouldn't ask to cite ideas from the extended evolutionary synthesis about different inheritance systems (as reviewed by Jablonka & Lamb, DOI: 10.1017/9781108685412) or the idea of inherency (Newman 2017, DOI: 10.1007/978-3-319-33038-9_78-1), but the authors may find them interesting. Same goes for our own work on mechanical morphogenesis which expands on the idea of a downward causality (Heuer and Toro 2019, DOI: 10.1016/j.plrev.2019.01.012)

Response 26

We thank the reviewer for recommending these papers, which we enjoyed reading and have deepened our thinking on the topic. In addition to toning down some of the language with respect to causality that our data cannot directly address, we have included additional discussion and references as follows:

In Discussion: "By establishing that some of these cortical zones are evident at the time of cortical folding, we lend support to a “protomap”(Rakic 1988; O'Leary 1989; O'Leary et al. 2007; Rakic et al. 2009) like model where the placement of some cortical folds is set-up by rapid tangential changes in cyto-laminar composition of the developing cortex(Ronan et al., 2014; Toro and Burnod, 2005; Van Essen, 2020). The DEMs are derived from fully folded adult donors, and therefore some of the measured genetic-folding alignment might also be induced by mechanical distortion of the tissue during folding(Llinares-Benadero and Borrell 2019; Heuer and Toro 2019). However, no data currently exist to conclusively assess the directionality of this gene-folding relationship.

Overall, the manuscript is very interesting and a great contribution. The amount of work involved is impressive, and the presentation of the results very clear. My comments indicate some aspects that could be made more clear, for example, providing additional methodological information in the supplemental material. Also, making aware the readers and future users of MAGICC of the methodological and conceptual challenges that remain to be addressed in the future for this field of research.
**Reviewer #2 (Recommendations For The Authors):**
Comment 1The supplementary figures seem to be missing from the eLife submission (although I was able to find them on europepmc)

Response 1

We apologize that these were not included in the documents sent to reviewers. The up-to-date supplementary figures are included in this resubmission and again on biorxiv.